# Vertical characterization of Highly Oxygenated Molecules (HOMs) below and above a boreal forest canopy

Qiaozhi Zha[1], Chao Yan[1], Heikki Junninen[1], Matthieu Riva[1], Nina Sarnela[1], Juho Aalto[1], Lauriane Quéléver[1], Simon Schallhart[1], Lubna Dada[1], Liine Heikkinen[1], Otso Peräkylä[1], Jun Zou[2], Clémence Rose[1], Yonghong Wang[1], Ivan Mammarella[3], Gabriel Katul[4,5], Timo Vesala[1], Douglas R. Worsnop[1,6], Markku Kulmala[1], Tuukka Petäjä[1], Federico Bianchi[1], and Mikael Ehn[1]

[1] Institute for Atmospheric and Earth System Research/Physics, Faculty of Science, University of Helsinki, P.O. Box 64, 00014 Helsinki, Finland
[2] CMA-NJU Joint Laboratory for Climate Prediction Studies, Institute for Climate and Global Change Research, School of Atmospheric Sciences, Nanjing University, Nanjing, China
[3] Department of Physics, University of Helsinki, P.O. Box 48, 00014 Finland
[4] Nicholas School of the Environment, Duke University, Durham, North Carolina, USA
[5] Department of Civil and Environmental Engineering, Duke University, Durham, North Carolina, USA
[6] Aerodyne Research, Inc., Billerica, MA 01821, USA

## 1 Abstract

While the role of highly oxygenated molecules (HOMs) in new particle formation (NPF) and secondary organic aerosol (SOA) formation is not in dispute, the interplay between HOM chemistry and atmospheric conditions continues to draw significant research attention. During the Influence of Biosphere-Atmosphere Interactions on the Reactive Nitrogen budget (IBAIRN) campaign in September 2016, profile measurements of neutral HOM molecules below and above the forest canopy were performed for the first time in the boreal forest SMEAR II station. The HOM concentrations and composition distributions below and above the canopy were similar during daytime, supporting a well-mixed boundary layer approximation. However, much lower nighttime HOM concentrations were frequently observed at ground level, which was likely due to the

formation of a shallow decoupled layer below the canopy. Near ground HOMs were
influenced by the changes in the precursors and oxidants, and enhancement of the loss
on surfaces in this layer, while the HOMs above the canopy top were not significantly
affected. Our findings clearly illustrate that near-ground HOM measurements
conducted under stably stratified conditions at this site might only be representative of
a small fraction of the entire nocturnal boundary layer. This could, in turn, influence the
growth of newly formed particles and SOA formation below the canopy where a large
majority of measurements are typically conducted.
**2    Introduction**
Highly oxygenated molecules (HOMs), a sub-group of the oxidation products of
volatile organic compounds (VOCs) identified by their high oxidation states, have been
recognized as important precursors for organic aerosol in the atmosphere (Ehn et al.,
2014). They have also been found to enhance new particle formation (NPF) and growth
(Kulmala et al., 2013; Zhao et al., 2013; Ehn et al., 2014; Bianchi et al., 2016; Kirkby
et al., 2016; Tröstl et al., 2016). The importance of HOMs has been confirmed in
ambient environments, especially in monoterpene-dominated regions such as the boreal
forest (Kulmala et al., 2013; Ehn et al., 2014), but also in high altitude mountain regions
(Bianchi et al., 2016) and in rural areas (Jokinen et al., 2014; Kürten et al., 2016). In
laboratory studies, HOM formation has been observed from various precursor
molecules (Ehn et al., 2017), including both biogenic and anthropogenic emissions.

The direct observation of HOMs has only recently become possible, following the
developments of the Atmospheric Pressure interface Time-Of-Flight (APi-TOF,
measures the naturally charged HOM) (Junninen et al., 2010) and Chemical Ionization
Atmospheric Pressure interface Time-Of-Flight (CI-APi-TOF, measures the neutral
HOM molecules) (Jokinen et al., 2012) mass spectrometers. Ehn et al. (2010) and
Bianchi et al. (2017) found that the naturally charged HOM clusters could be observed
every night in the boreal forest during spring. Out of the observed ambient mass spectra,
a significant part could be reproduced in a chamber by introducing the monoterpene $\alpha$-
pinene ($C_{10}H_{16}$, the major biogenic VOC in the boreal forest) and ozone ($O_3$) (Ehn et
al., 2012).

Further investigations of HOM formation chemistry have been done in both laboratory
and field studies. Based on current understanding from laboratory experiments, the
formation of HOM molecules involves three main steps: 1) initial formation of peroxy
radicals ($RO_2$) from VOC oxidation; 2) $RO_2$ autoxidation, that is, the isomerization of
the $RO_2$ via intramolecular H-shifts and subsequent oxygen ($O_2$) additions; and 3)
radical termination, forming closed-shell molecules (Crounse et al., 2013; Ehn et al.,
2014; Jokinen et al., 2014, 2016; Rissanen et al., 2014; Mentel et al., 2015). In the
atmosphere, HOM formation studies are complicated by the plethora of different
compounds and processes taking place. However, recent ambient measurements
together with factor analysis were able to shed light on the HOM formation pathways
in the boreal forest (Yan et al., 2016). They showed that the majority of the daytime
production of HOMs was from reactions initiated by the oxidation of monoterpenes
(MT) with hydroxyl radical (OH) or $O_3$. The $RO_2$ after autoxidation were often
terminated by hydroperoxyl radicals ($HO_2$) or self-termination (Orlando and Tyndall,
2012), to form a non-nitrate HOM monomer ($CHO_{monomer}$, mainly $C_9$ and $C_{10}$
compounds, with masses between 290-450 Th after clustering with the charging ion
($NO_3^-$) of the instrument), or reacting with nitrogen oxides ($NO_x = NO + NO_2$) to form
organonitrate HOM monomers ($CHON_{monomer}$). During nighttime, MT were mainly
oxidized by $O_3$ and $NO_3$ radicals. Furthermore, due to the lower nocturnal $HO_2$ and
$NO_x$ concentrations, besides the production of $CHON_{monomer}$, the $RO_2$ products readily
reacted with other $RO_2$ to form either non-nitrate HOM dimers ($CHO_{dimer}$, mainly $C_{16-}$
$_{20}$ compounds with masses between 450-600 Th after clustering with $NO_3^-$) or
organonitrate HOM dimers ($CHON_{dimer}$), depending on the oxidants forming the $RO_2$
radical. (Ehn et al., 2014; Jokinen et al., 2014; Yan et al., 2016; Berndt et al., 2018).

Beyond those chemical pathways, varied meteorological conditions are also factors
influencing the MT and oxidants at different heights above the forest floor.
Unsurprisingly, the oxidants producing HOMs (e.g. $O_3$) have been found almost
uniformly distributed within the well-mixed daytime boundary layer (Chen et al., 2018).
In contrast, the nocturnal boundary layer was shallow with stability regimes that
depended on radiative cooling within the canopy and turbulent shear stresses at the
canopy top. In Hyytiälä, the depletion of $O_3$ below the canopy has been frequently
observed during nighttime, while the $O_3$ above the canopy was less affected (Chen et
al., 2018). The MT concentration at ground level increased when $O_3$ was depleted
(Eerdekens et al., 2009). The inhomogeneous distribution of the precursors and
oxidants below and above the canopy might further impact nocturnal HOM
distributions, which frames the scope of this study. Until now, all CI-APi-TOF
deployments have been at ground level, and the main subject of inquiry here is the
vertical information on HOMs and the role of meteorological condition in shaping them.
A characterization of the HOMs at different heights provides a decisive advantage in
disentangling the role of non-uniform mixing within the atmospheric layers impacted
by strong thermal stratification, especially inside the canopy volume.

The first measurements of the HOM concentrations at two different heights (36 m and
1.5 m a.g.l.) are presented and discussed. The influence of boundary layer dynamics on
the HOMs at these different heights at SMEAR II station are analyzed and characterized
in conjunction with auxiliary turbulence and micrometeorological measurements.
**3   Experimental**
**3.1  Measurement site description**
The measurements were performed at the SMEAR II station (Station for Measuring

Ecosystem–Atmosphere Relations) in the boreal forest in Hyytiälä, southern Finland (61°51' N, 24°17' E, 181 m a.s.l., Hari and Kulmala, 2005; Hari et al., 2013) during September 2016. There is no large anthropogenic emission source at or near the site. The closest sources are the two sawmills ~5 km southeast of the site, and from the city area of Tampere (~60 km away). The forest surrounding the station is primarily Scots pine with a mean canopy height of ~17.5 m, a total leaf area index (LAI) of ~6.5 $m^2m^{-2}$, a stand density of ~1400 trees $ha^{-1}$, and an average diameter at breast height (DBH) of ~0.16 m (Bäck et al., 2012; Launiainen et al., 2013). The forest floor is majorly covered with a shallow dwarf shrub (a LAI of ~0.5 $m^2m^{-2}$) and moss layer (a LAI of ~1 $m^2m^{-2}$) (Kulmala et al., 2008; Launiainen et al., 2013). The planetary boundary layer height at the SMEAR II station has been determined from previous studies using radiosondes (Lauros et al., 2007; Ouwersloot et al., 2012) and balloon soundings (Eerdekens et al., 2009). Roughly, these heights span some 400 m (March) to 1700 m (August) at noontime, and 100 m (March) to <160 m (April) at midnight.

**3.2 Instrumentation**

Concentration of HOM molecules were measured with two nitrate-ion based CI-APi-TOF mass spectrometers. The CI-APi-TOF measuring at higher altitude was deployed at the top of a 35 m tower located ~20 m horizontally from the ground measurement location. Both instruments were working in rooms with air-conditioning and room temperatures controlled at 25 °C. The inlets of the two instruments were pointed to the southeast direction and fixed at ~36 m and ~1.5 m above ground. The tower measurement is at about twice the canopy height, which is still within the roughness sublayer of the forest (Raupach and Thom, 1981). The instrument setup of the two CI-APi-TOF mass spectrometers were similar. In brief, the CI-APi-TOF was the combination of a chemical ionization (CI) inlet, and an atmospheric pressure interface time-of-flight (APi-TOF) mass spectrometer (Aerodyne Research Inc., USA, and Tofwerk AG, Switzerland). The ambient air was first drawn into the inlet with a sample

flow of 7 lpm (liter per minute), and then centered to an ion reaction tube surrounded
by sheath flow (filtered air, 35 lpm). Meanwhile, the nitrate ions carried by the sheath
gas, which were generated by exposing the nitric acid ($HNO_3$) to soft x-ray radiation,
were guided into the sample gas by an electrical field at ambient pressure (~100 ms
reaction time). Neutral molecules (M) in the sample air were ionized by either
clustering with charged nitrate/nitric acid ($(HNO_3)_{n=0-2} \cdot NO_3^-$) to form $(M) \cdot NO_3^-$ cluster
ions, or losing a proton to the charging ions to form deprotonated ions (e.g.,
$H_2SO_4 + NO_3^- \rightarrow HSO_4^- + HNO_3$). The ions then entered the APi part, which was a three-
stage vacuum chamber, through a pinhole. In the APi, two quadrupoles and a stack of
ion lenses guided the ions into the TOF mass analyzer, where ions were separated based
on their mass-to-charge ($m/z$) ratios. A more detailed description of this instrument has
been given by Junninen et al. (2010) and Jokinen et al. (2012), and discussion on
selectivity of this nitrate ion charging by Hyttinen et al. (2015). Mass spectra obtained
from the instrument were analyzed using the 'tofTools' program described in Junninen
et al. (2010). Determination of the concentration of a measured molecule M was based
on the following equation:
$$[M] = \frac{\sum M}{\sum reagent\ ion\ count\ rates} \times C \qquad\qquad\qquad (1)$$
where the sum of ion count rates ($\sum M$) in the numerator includes all detected ions
relating to compound $M$, whether deprotonated or in clusters with reagent ions, and the
sum of reagent ion count rates in the denominator is the total signal of the nitrate ions.
$C$ is the calibration coefficient, which was assigned the same value for all detected
compounds. This assignment is only valid for compounds that cluster with the reagent
ions at the collision limit, such as $H_2SO_4$ (Viggiano et al., 1997) and have equal collision
rates. The collision rates of nitrate ions with $H_2SO_4$ and with HOMs are expected to be
very close (Ehn et al., 2014). Here, a calibration coefficient of $1 \times 10^{10}$ molec cm$^{-3}$,
estimated from previous calibrations with similar settings using sulfuric acid and
theoretical constraints (Ehn et al., 2014), with an uncertainty of at least -50%/+100%,
was used in calculating the HOM concentrations for both instruments. Ultimately, the
absolute HOM concentrations in this work are of secondary importance, as we focus on
the relative comparison of HOM concentrations measured at different heights. However,
the comparability of the two CI-APi-TOF instruments is of great importance, and
results cannot be allowed to vary e.g. as a result of inevitable differences in the mass-
dependent transmission efficiency (TE). For a detailed discussion on factors affecting
the TE of a CI-APi-TOF, we refer to Heinritzi et al. (2016). To this end, instead of
directly evaluating the TE of each instrument, a "relative" TE of the two CI-APi-TOFs
was used for data correction: we selected a time period at noon-time on September 9
with a well-mixed boundary layer, identified with the clear and sunny weather and
homogeneous vertical distribution of monoterpene and other trace gases, and assumed
the HOM concentrations at the two heights to be the same. Thus, the relative TE was
obtained from the concentration ratio between the two CI-APi-TOFs at each $m/z$ (Figure
1). A fitted relative TE curve ($R^2 = 0.97$), which represents how the TE of the tower CI-
APi-TOF was changed at each m/z over the TE of the ground one, was obtained using
power law regression. Weaker correlation was obtained in the 200-250 and 500-600 Th
mass ranges, but in the mass range where most of the HOMs were located (290-500 Th)
there is very little scatter around the fitted curve, clearly suggesting that observed
differences in the two instruments responses were mainly due to differences in TE. To
test our assumption of negligible vertical gradients of HOMs during daytime, we
analyzed the behavior of sulfuric acid. We found that the uncertainty related to this
assumption corresponds to a value of 26% (see Figure S1). An upper limit of uncertainty
relating to our TE correction (Figure 1) was also estimated, yielding a value of 10%,
giving a total uncertainty from these two sources of 28%. This value is much smaller
than the observed deviation of HOM concentrations during inversion nights, as will be
discussed later. Additionally, an inter-comparison between the two instruments with a
permeation tube containing trinitrotriazinane ($C_3H_6N_6O_6$) was conducted in the field
right after the campaign. The results showed good agreement with the relative TE,
lending confidence to the method used here. Finally, it should be noted that the
difference in TE between the two instruments was larger than one would normally
expect, since the tower CI-APi-TOF had been tuned for higher sensitivity at the largest
masses (at the expense of transmission at the lower masses).

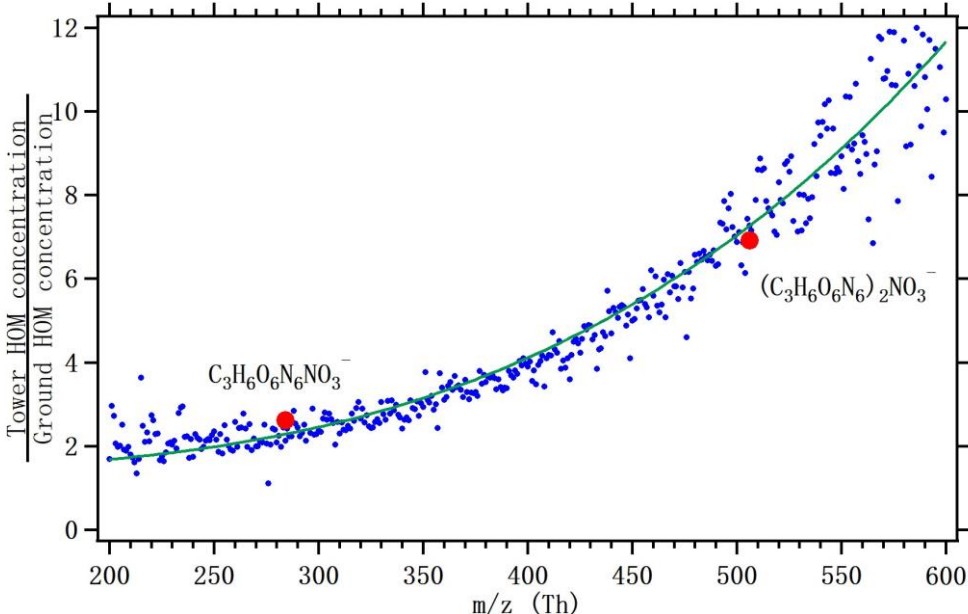


Figure 1. The relative transmission curve between the two CI-APi-TOF mass spectrometers, determined
during a period of strong turbulent mixing. Inter-comparison results using a permeation tube containing
trinitrotriazinane ($C_3H_6N_6O_6$) are shown in red circles. The fitted green line was used to scale the
measured signals between the two instruments to match, in order to compare relative changes during
times of limited vertical mixing.

In comparison to the direct determination of TE (Heinritzi et al., 2016), this method
increases the uncertainty in the quantification of HOM concentrations. However, as
mentioned, a more accurate knowledge of the exact HOM concentrations would not
influence the main findings of this study.

The MT, trace gases, and meteorological parameters were continuously monitored at
the different heights (4.2 m, 8.4 m, 16.8 m, 33.6 m, 50.4 m, 67.2 m, 101m, and 125 m)
on a 126 m mast ~100 m away from the location of the CI-APi-TOFs. The data at 4.2
m and 33.6 m were used in this study to represent the concentrations at near ground and
tower level, respectively. MT concentrations were measured every third hour using a
proton transfer reaction mass spectrometer with a lower detection limit of 1 pptv (PTR-
MS, Ionicon Analytik GmbH; Taipale et al., 2008). The $O_3$ concentration was measured
with an UV light absorption analyzer that had a lower detection limit of 1 ppbv (TEI
model 49C, Thermo Fisher Scientific, USA). The $NO_x$ measurement was conducted
using a chemiluminescence analyzer (TEI model 42C TL, Thermo Fisher Scientific,
USA). The lower detection limit of the $NO_x$ analyzer is 100 pptv. The $CO_2$ measurement
was performed using an infrared detection system (LI-840, LiCor Biosciences, Lincoln,
NE, USA). The aerosol number concentration size distributions were obtained with a
twin differential mobility particle sizer (twin-DMPS) for the size range from 3-1000
nm (Aalto et al., 2001) at 8 m height above ground, and was used to calculate
condensation sink (CS) based on the method from Kulmala et al. (2001). Air
temperature was measured with PT-100 resistance thermometers. Air relative humidity
(RH) was measured with RH sensors (Rotronic Hygromet model MP102H with
Hygroclip HC2-S3, Rotronic AG, Switzerland). Global radiation (solar radiation in
wavelength range of 0.3-4.8 μm) was obtained with a Pyranometer (Reemann TP3,
Astrodata, Estonia) above the canopy top at 18 m. All the data presented are at 10 min
averaging intervals, except for the MT (in 1-hour averaging interval). A schematic
figure showing sampling locations of all the measured parameters is provided in Figure
S2.

**4   Results and discussion**
**4.1  Data overview**
The Influence of Biosphere-Atmosphere Interactions on the Reactive Nitrogen budget
(IBAIRN) campaign was conducted from September 1 to 25, 2016. After data quality
checks, only the measurements collected after September 5 were used. Figure 2 shows
the overall time series of the meteorological parameters measured at ground and tower
levels, including the temperature, RH, global radiation, concentrations of trace gases,
MT, and total HOMs. The weather was generally sunny and clear during the campaign
except for a few cloudy (September 10, 15, and 22-23) and drizzling (September 24
and 25) days. The mean air temperature and RH observed at ground level were $10.8 \pm$
$3.3\ °C$ and $87 \pm 13\ \%$ ($1\sigma$ standard deviation), and at the tower level were $10.5 \pm 3.0\ °C$
and $88 \pm 14\ \%$, respectively. The $O_3$ concentrations measured at ground and tower levels
were $21 \pm 8$ ppbv and $25 \pm 6$ ppbv, respectively. The air temperature, RH and $O_3$
measured at the two heights were close to each other during daytime. The $NO_x$
concentrations were quite low throughout the campaign, the mean $NO_x$ concentrations
were mostly around the reported detection limit at $0.4 \pm 0.4$ ppbv (ground) and $0.4 \pm$
$0.5$ ppbv (tower), yet showed an overall good agreement between the measurements at
the different heights. The MT concentrations at ground level ($0.38 \pm 0.34$ ppbv on
average) were generally higher than that above the canopy level ($0.20 \pm 0.16$ ppbv).

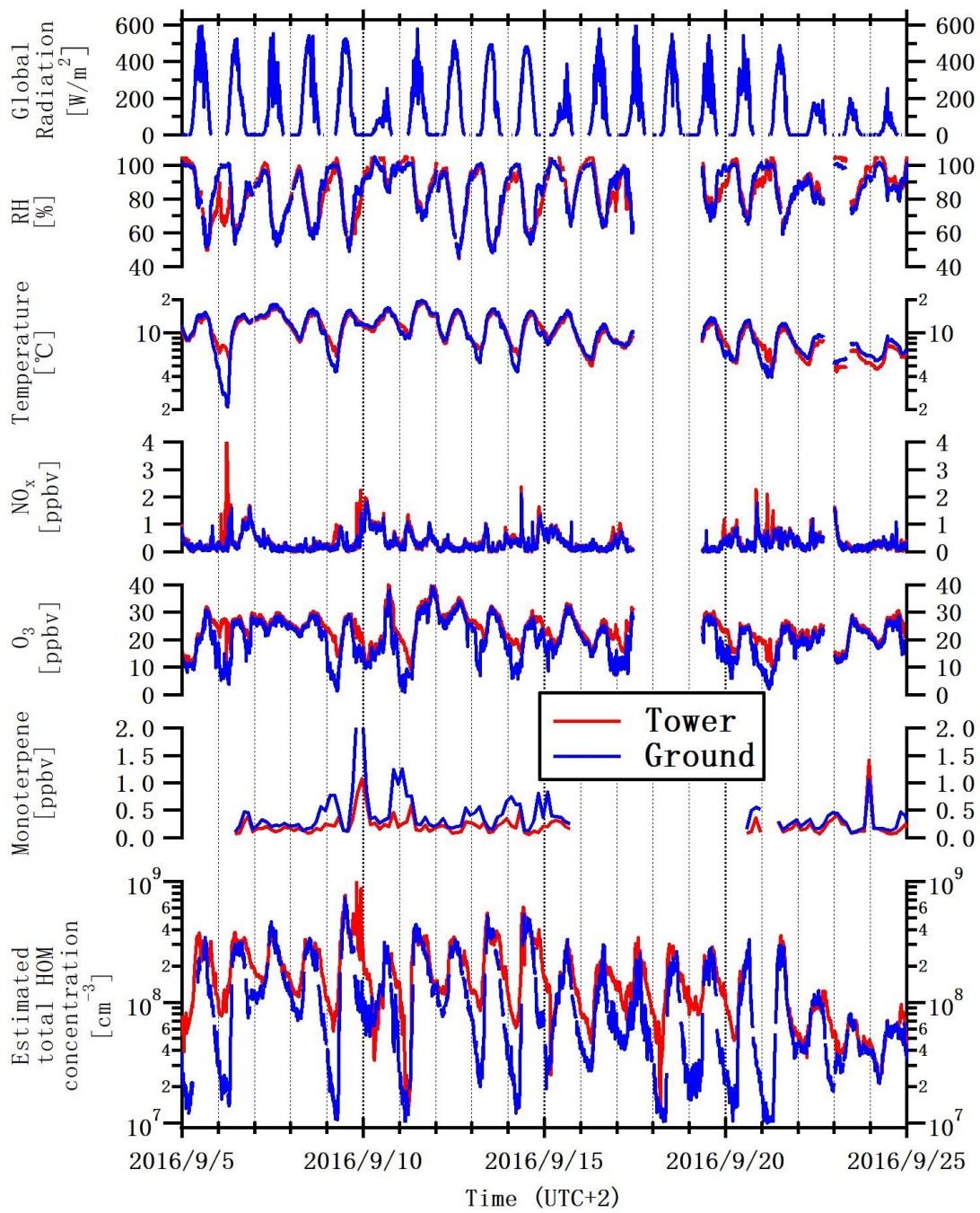


Figure 2. The overall time series of the measured trace gases, meteorological parameters and estimated
total HOM concentrations at the ground (blue) and tower (red) levels.

The estimated total HOM concentration is representative for the overall concentration
level of HOMs, and is here defined as the sum of the detected signals between ions
from *m/z* 200 to 600 after removing identified background peaks. The gaps in the
estimated total HOM at ground level were due to automatic zero-checks. During the
campaign, a significant difference was found in the estimated total HOM concentrations
below and above the canopy (mean and median concentrations of $1.1 \pm 1.7 \times 10^8$ cm$^{-3}$
and $0.8 \times 10^8$ cm$^{-3}$ at ground level, $1.7 \pm 1.3 \times 10^8$ cm$^{-3}$ and $1.3 \times 10^8$ cm$^{-3}$ at tower
level). The causes of these differences (~ 55% in mean and ~71% in median) frame the
upcoming discussion.

**4.2 Inter-comparison of estimated total HOM concentrations**
The estimated total HOM concentrations at the two heights were not different during
the day (mean $\pm 1\sigma$ standard deviation and median concentrations of $4.1 \pm 2.3 \times 10^8$ cm$^{-}$
$^3$ and $3.6 \times 10^8$ cm$^{-3}$ at ground level, $4.3 \pm 2.6 \times 10^8$ cm$^{-3}$ and $4.0 \times 10^8$ cm$^{-3}$ at tower
level), which validates the use of only one day of data for scaling the TE of the ground
CI-APi-TOF to match the HOM signals of the two instruments. The good daytime
agreement throughout the campaign period also verifies that the response of each
instrument stayed stable. Contrary to the daytime results, the estimated total HOM
concentration at ground level usually diverged from the tower measurement in the
nocturnal boundary layer. The concentration below the canopy became even lower
when temperature inversions were observed, accompanied by a decreasing ground-
level $O_3$ and increasing MT concentrations. Figure 3 shows a comparison between the
estimated total HOM concentrations observed at two heights. Herein, good agreement
could be found for the group of points representing the concentrations around noontime
($R^2 = 0.89$). The points indicating the nighttime estimated total HOM concentrations
were scattered ($R^2 = 0.28$), and the ground concentrations were found to be much lower
than the tower ones.

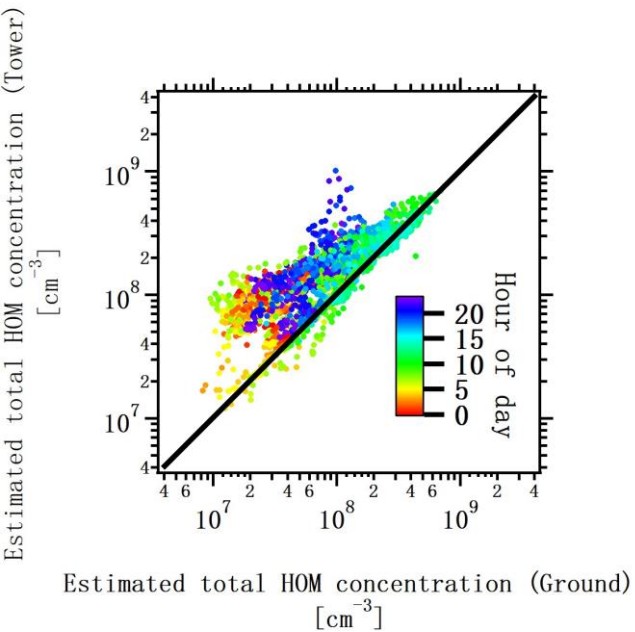

Figure 3. Comparison between ground (x-axis) and tower (y-axis) measurements of the estimated total HOM concentrations. The black line denotes the 1:1 ratio. Color code indicates the sampling time of HOMs.

Figure 4 shows the mean mass spectra (in unit mass resolution, UMR, for *m/z* 200 – 600) obtained from the ground and tower. It is worth mentioning that there might be some signals not attributable to HOMs in the plotted spectra, but only in little proportion. Only selected periods (09:00-15:00 for daytime and 21:00-03:00 for nighttime, local winter time (UTC +2)) are included in the averaging period to eliminate the effect of sunrise and sunset periods. During daytime, a good agreement ($R^2 = 0.87$) was obtained from the mass-by-mass comparison using the UMR concentrations extracted from daytime mean spectra, suggesting a uniform composition distribution in the daytime boundary layer condition. During nighttime, the mean concentrations of all HOM molecules in the ground mean spectra were much lower than the tower spectra. The HOM concentrations shown in the ground and tower mean spectra were also less correlated. Therefore, a logical outcome is that the conditions below and above the canopy are experiencing different turbulent mixing strength and/or source-sink regimes

during night.

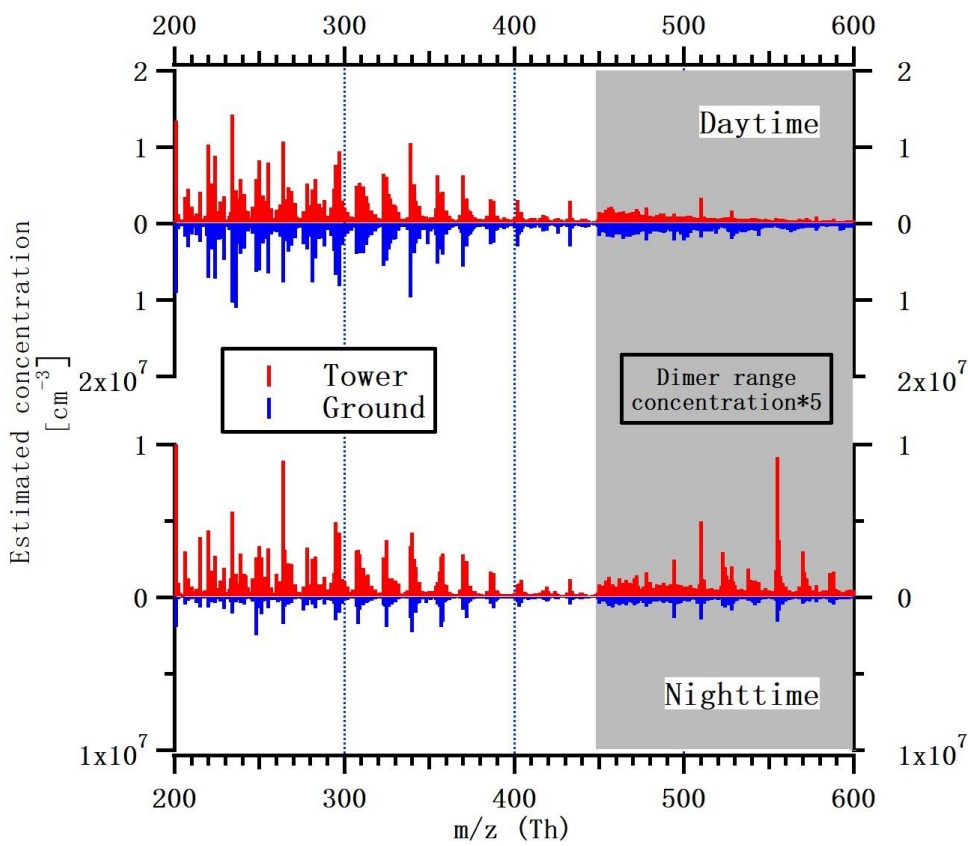

Figure 4. Mean mass spectra with the averaging periods of daytime (09:00-15:00) and nighttime
(21:00-03:00) at ground and tower levels.

### 4.3 Influence of nocturnal boundary layer dynamics and micrometeorological processes

The nighttime HOMs at ground level are likely influenced by transport processes below
the canopy, since the estimated total HOM concentrations were found much lower in
the nights when temperature inversions were observed. To further investigate the
potential impact of such micrometeorological phenomena on ground level HOMs, for
the nights during the campaign without precipitation or instrument failure, were
selected (14 nights in total) and categorized into 2 types based on the occurrence of
temperature inversions: 1) the "non-inversion night" type included 7 nights when no
temperature inversion was recorded; 2) the "inversion night" type category consisted
of 7 nights that had encountered temperature inversions, and the ground temperatures
were generally ~1 °C lower than tower temperatures during these nights.

**4.3.1 Statistics of the "non-inversion night" and "inversion night" types**
Table 1 shows the overall statistics including the mean and median values of the
temperatures, $O_3$, $NO_x$, MT and estimated total HOM concentrations for the "non-
inversion night" and "inversion night" types. In the non-inversion nights, the air below
and above the canopy was relatively well-mixed. The mean and median concentrations
of the ground $O_3$ ($21 \pm 8$ ppbv and 22 ppbv) were close to the tower values ($25 \pm 6$ ppbv
and 24 ppbv). The slight difference might be attributed to the higher VOC emissions
(Rantala et al., 2014) and larger sink near ground level. In contrast, during the inversion
nights, the mean estimated total HOM concentration and $O_3$ at ground level were
generally much lower, only ~33% and ~69% of the tower concentrations, respectively.
Instead, the mean and median ground MT concentration ($0.70 \pm 0.28$ ppbv and 0.70
ppbv) were ~3 times higher than the tower ones ($0.24 \pm 0.04$ ppbv and 0.23 ppbv),
respectively. The measured $NO_x$ levels were similar in both categories and heights,
though the ambient concentrations were close to the detection limit and therefore small
differences might not be observable.

**4.3.2 Case study**
Two individual nights representing the "non-inversion night" and "inversion night"
types were selected and further compared. Figure 5a shows the time series of the
meteorological parameters, trace gases and HOMs measured at ground and tower
levels of one selected night of "non-inversion night" type (September 11-12, from 21:00
to 03:00). A number of measures can be used to assess the local atmospheric stability
conditions at a given layer. These measures are commonly based on either the Obukhov
length and its associated atmospheric stability parameter or a Richardson number (flux-
based, gradient-based, or bulk). Because of its simplicity and the availability of high
resolution mean air temperature profiles, the bulk Richardson number ($Ri$) was used
here (Mahrt et al., 2001; Mammarella et al., 2007; Vickers et al., 2012; Alekseychik et
al., 2013). It is calculated using:
$Ri = \frac{g \Delta \overline{\theta} \Delta z}{\overline{\theta} (\overline{u})^2}$                                                (2)
where $g$ is the gravitational acceleration, $\Delta \overline{\theta}$ and $\Delta z$ are the mean potential
temperature (10 min averaging interval, same as measurement data) and height
difference between the ground and tower levels, respectively, $\overline{\theta}$ and $\overline{u}$ are the mean
potential temperature and mean wind velocity at tower level, respectively. During the
selected "non-inversion" night, $Ri$ was generally positive but close to 0 (shown in
Figure 5a), indicating a weakly stable and relatively well-mixed (i.e. $\Delta \overline{\theta} \to 0$)
condition (Mahrt, 1998; Mammarella et al., 2007). This was also confirmed using the
well correlated ground and tower MT and trace gases concentrations.

**Table 1**. Summary of the "Non-inversion night" and "Inversion night" types.

| Type | | Non-inversion night | | | | | Inversion night | | | | |
|---|---|---|---|---|---|---|---|---|---|---|---|
| **Date** | | September 6, 7, 9, 11, 15, 16, 21[*] | | | | | September 5, 8, 10, 12, 13, 14, 19[**] | | | | |
| **Parameters** | | Temperature [°C] | $O_3$ [ppbv] | $NO_x$ [ppbv] | MT [ppbv] | Estimated total HOM [$10^8$ cm$^{-3}$] | Temperature [°C] | $O_3$ [ppbv] | $NO_x$ [ppbv] | MT [ppbv] | Estimated total HOM [$10^8$ cm$^{-3}$] |
| **Tower** | Mean ± 1σ standard deviation | 10.2 ± 2.6 | 25 ± 6 | 0.5 ± 0.5 | 0.31 ± 0.31 | 2.9 ± 1.9 | 9.5 ± 1.7 | 24 ± 2 | 0.4 ± 0.3 | 0.24 ± 0.04 | 2.4 ± 0.8 |
| | Median | 10.9 | 24 | 0.4 | 0.17 | 2.8 | 9.2 | 23 | 0.3 | 0.23 | 2.3 |
| **Ground** | Mean ± 1σ standard deviation | 10.6 ± 2.7 | 21 ± 8 | 0.4 ± 0.4 | 0.52 ± 0.74 | 1.6 ± 0.6 | 8.3 ± 2.2 | 16 ± 6 | 0.3 ± 0.2 | 0.70 ± 0.28 | 0.8 ± 0.4 |
| | Median | 11.5 | 22 | 0.3 | 0.22 | 1.7 | 8.5 | 17 | 0.3 | 0.70 | 0.7 |

[*]MT data not available on September 5 and 19.
[**]MT data not available on September 15 and 16

Selected HOM molecules representing the major HOM types (and formation pathways) were summed up and categorized into 4 groups, as shown in Table 2. Each pathway might be influenced differently by boundary layer dynamics and micrometeorological processes. In this study, OH-initiated HOMs were assumed negligible due to the very low OH level in the nocturnal boundary layer.

**Table 2**. Compositions of selected HOM molecules and their main oxidants (Yan et al., 2016).

| | Molecule compositions | Main oxidants | Main terminators |
|---|---|---|---|
| **$CHO_{monomer}$** | $C_{10}H_{14}O_7$, $C_{10}H_{14}O_9$ | $O_3$ | Self-terminate or $RO_2$ |
| **$CHON_{monomer}$** | $C_{10}H_{15}O_9N$, $C_{10}H_{15}O_{11}N$ | $O_3$ or $NO_3$ | NO or self-termination/$RO_2$ |
| **$CHO_{dimer}$** | $C_{19}H_{28}O_{11}$, $C_{20}H_{30}O_{14}$ | $O_3$ | $RO_2$ |
| **$CHON_{dimer}$** | $C_{20}H_{32}O_{12}N_2$, $C_{20}H_{31}O_{13}N$ | $NO_3$ | $RO_2$ |

All the HOM groups in Figure 5a show stable patterns, and good agreement is observed between the ground and tower measurements in the first half of the night. Variations were observed when air mass change occurred at around 01:00, as indicated by the drop of $NO_x$ concentration and horizontal wind shift (not shown here). A rapid decrease was found in CS, which represents the rate of condensation of low-volatile vapors onto the existing aerosol particles (Dada et al., 2017), implying that the aerosol population also changed. However, the HOM groups were still well-correlated with each other, suggesting well-mixed conditions in the non-inversion night.

Figure 5b shows the time series of the trace gases, MT, and HOM groups of both ground and tower measurements during an "inversion night" case (September 8-9, from 21:00 to 03:00). *Ri* was generally higher during this night, and increased from ~0.03 (indicating weakly stable condition, Mammarella et al., 2007), at around midnight, to a maximum of ~1.13 (indicating very stable condition) in the remaining night period.

Roughly, $Ri$ values in excess of unity indicate that stably stratified conditions appreciably diminish the inverse turbulent Prandtl number ($Pr$) and the efficiency of turbulence to mix heat when compared to momentum (Katul et al., 2014). The parameters measured at tower level were not significantly affected by strong $Ri$ fluctuations throughout the night, in contrast, significant variations were observed at ground level.

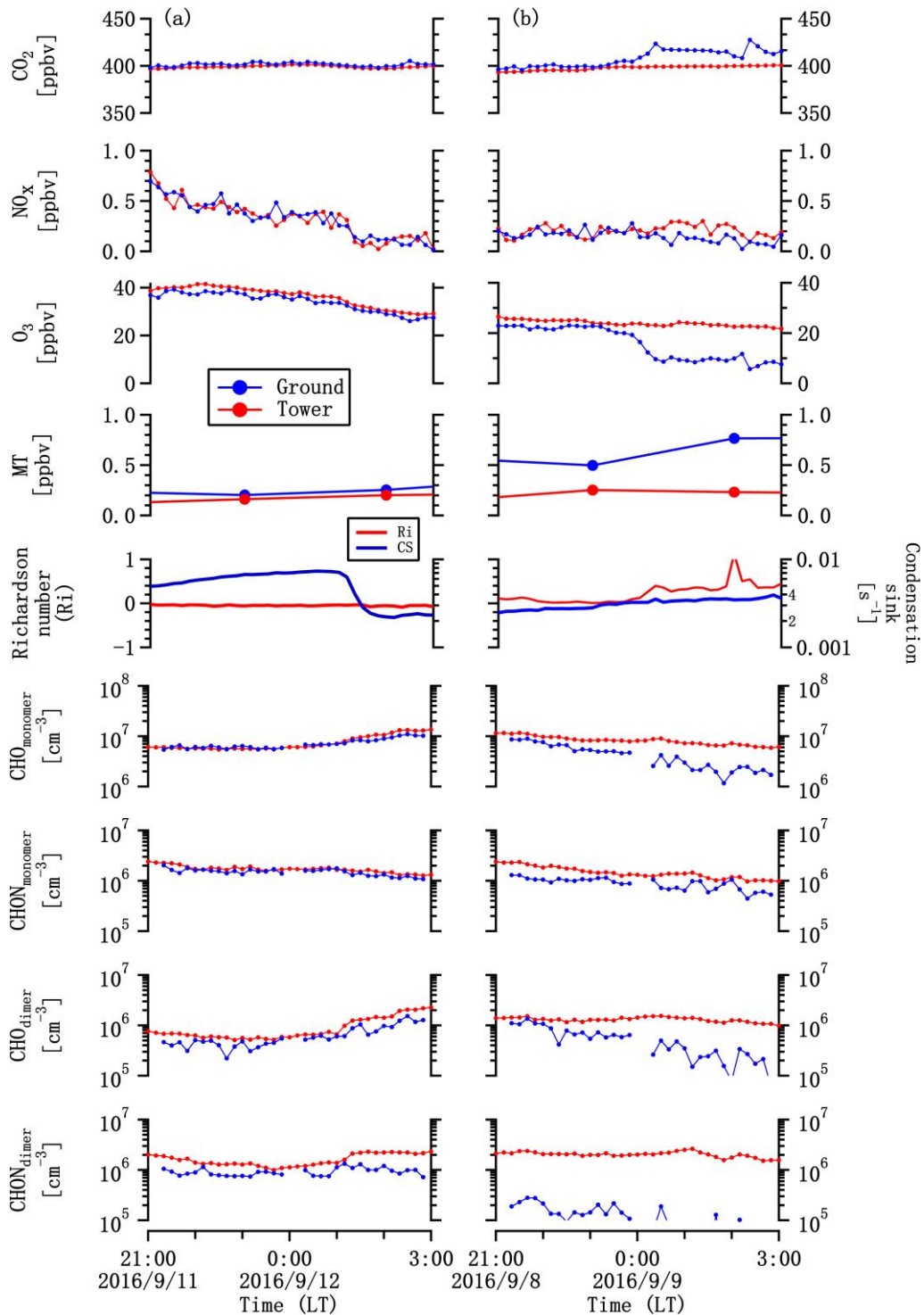

Figure 5. (a) Time series of ground and tower concentrations of $CO_2$, $NO_x$, $O_3$, MT, and selected HOM groups in the selected "non-inversion night" (September 11), and (b) "inversion night" (September 8). *Ri* is calculated with the meteorology data of ground and tower levels. CS is determined based on the aerosol data measured at 8 m above ground level.

The ground $O_3$ concentration experienced a rapid decrease at midnight. In about an hour

(from 23:30-00:30), ground $O_3$ concentration dropped by more than half (from 20 ppbv to 9 ppbv), and $CO_2$ concentration increased as well (from 404 ppbv to 423 ppbv). To the contrary, the MT concentration at ground level was almost doubled (from 0.49 ppbv to 0.80 ppbv) during the same period. Theoretically, the enhancement of HOM precursor VOC and the decrease of oxidant would compensate each other if the sink remained the same, and the ground HOM concentrations should also keep constant. However, all the HOM groups showed significant decrease after midnight, despite the CS (generally the main sink for HOM in the atmosphere) staying practically constant. In particular, the concentration of the $CHO_{monomer}$ group dropped ~80%, from $8.6 \times 10^6$ $cm^{-3}$ to $1.7 \times 10^6$ $cm^{-3}$, and the concentration of the $CHO_{dimer}$ group decreased from $1.5 \times 10^6$ $cm^{-3}$ to ~$0.1 \times 10^6$ $cm^{-3}$. The concentrations of the $CHON_{monomer}$ and $CHON_{dimer}$ groups also experienced large declines (~34% and ~50%, respectively), in the latter half of the night. At 03:00, the $CHON_{dimer}$ concentration was already below the detection limit ($1 \times 10^4$ $cm^{-3}$). Therefore, the much lower ground HOM concentrations might not be totally explained by the change of HOM production, but also due to some other processes such as additional losses.

A previous study by Alekseychik et al. (2013) at SMEAR II station showed that nocturnal decoupled air layers were frequently (with a fraction of 18.6% based on a long-term dataset) observed under high $Ri$ conditions in the boreal forest. The decoupled layer could strongly influence the ground $O_3$, MT, and $CO_2$ concentrations (Rannik et al., 2009, 2012; Alekseychik et al., 2013; Chen et al., 2018), and could also explain the occurrence of the strong temperature inversion during the inversion nights. To explore the possible mechanism resulting in significantly different $O_3$, MT and HOM concentrations below the canopy, the mean continuity equation for high Reynolds number flows within the canopy is formulated as (e.g. Katul et al. 2006):

$$\frac{\partial \bar{c}}{\partial t} + \bar{U}\frac{\partial \bar{c}}{\partial x} + \bar{W}\frac{\partial \bar{c}}{\partial z} = -S - \frac{\partial \overline{w'c'}}{\partial z} - \frac{\partial \overline{u'c'}}{\partial x} \tag{3}$$

$$N_1 + N_2 + N_3 = N_4 + N_5 + N_6 \tag{4}$$

where $t$ is time, $x$ and $z$ are the longitudinal and vertical directions, respectively, $C$

is the scalar concentration, $U$ and $W$ are the longitudinal and vertical velocity components, $\overline{w'c'}$ and $\overline{u'c'}$ are the turbulent scalar fluxes in the vertical and horizontal, respectively, and $S$ represents the net sources or sinks (physical, chemical, and biological) of $C$, and overline represents time averaging over turbulent scales. The 6 terms in this equation represent the following (left to right): local rate of change($= N_1$), horizontal advection by the mean velocity $(= N_2)$, vertical advection by the mean velocity $(= N_3)$, net sources or sinks $(= N_4)$, net vertical transport by the vertical turbulent flux gradient $(= N_5)$, net horizontal transport by the horizontal turbulent flux gradient $(= N_6)$. Generally, $|N_6| \ll |N_5|$, and is hereafter ignored in the discussion.

During the non-inversion night, the ground $O_3$ could be replenished either by vertical turbulent transport $(N_5)$, mean vertical advection from upper boundary layer $(N_3)$, or horizontal advection below the canopy $(N_2)$ (as shown in Figure 6). However, for highly stratified flows, $N_5$ becomes small, as the efficiency of turbulence to transport $O_3$ to layers near the ground becomes weak (Katul et al., 2014). Vertical and horizontal advection were also small within such a stable layer, and the reduced mean velocity would result in smaller contributions from $N_2$ and $N_3$. Note that these advective terms tend to be opposite in sign by the virtue of the mean fluid continuity equation (Katul et al., 2006). Instead, the sink of $O_3$ $(N_4)$ was stronger because of the increasing loss due to a higher surface area-to-volume density (S/V) in this shallow decoupled layer. Under this circumstance, the ground $O_3$ concentration dramatically decreased when the air layer was forming, and eventually reached a much lower concentration. The decoupled layer also affected MT and $CO_2$ below the canopy in the inversion night, but resulted in concentration increases as opposed to $O_3$. The weakened vertical turbulence $(N_5)$ tended to retain the emissions from ground and understory vegetation within the layer, though $N_4$ also increased. In general, the increased $CO_2$ (primary source from the ground) and MT (primary source from the canopy) at ground level are good indicators for the extent of the mixing in the shallow decoupled layer. At the same time, the strong decrease of $O_3$ shows how the sinks in this layer are no longer balanced by a large flux

of $O_3$ from upper layers. However, the stabilization of ground-level $O_3$ concentrations at non-zero values after the initial fast decrease suggests that a small amount of inflow, either via $N_3$ or $N_5$, is still taking place.

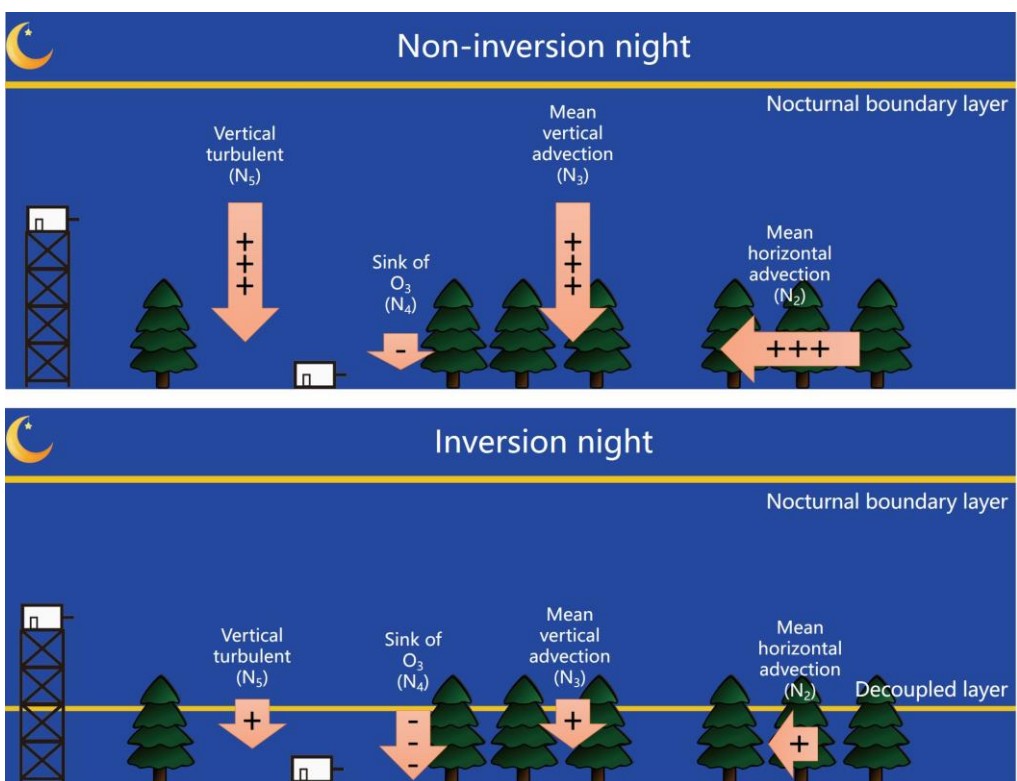

Figure 6. Schematic figure showing how vertical mixing, vertical advection, and horizontal advection influence ground level $O_3$ concentrations differently in non-inversion nights and inversion nights at SMEAR II station.

Therefore, the differences between the ground and tower measurements were due to the joint effects of: (i) decoupling between the stably stratified near-ground layer and the canopy top, and the consequent formation of a shallow layer, (ii) weakening of advective and turbulent flux transport terms thereby inhibiting mass exchange between the ground decoupled layer and the remaining nocturnal boundary layer, and (iii) increased surface area to volume within the decoupled layer thereby enhancing $N_4$.

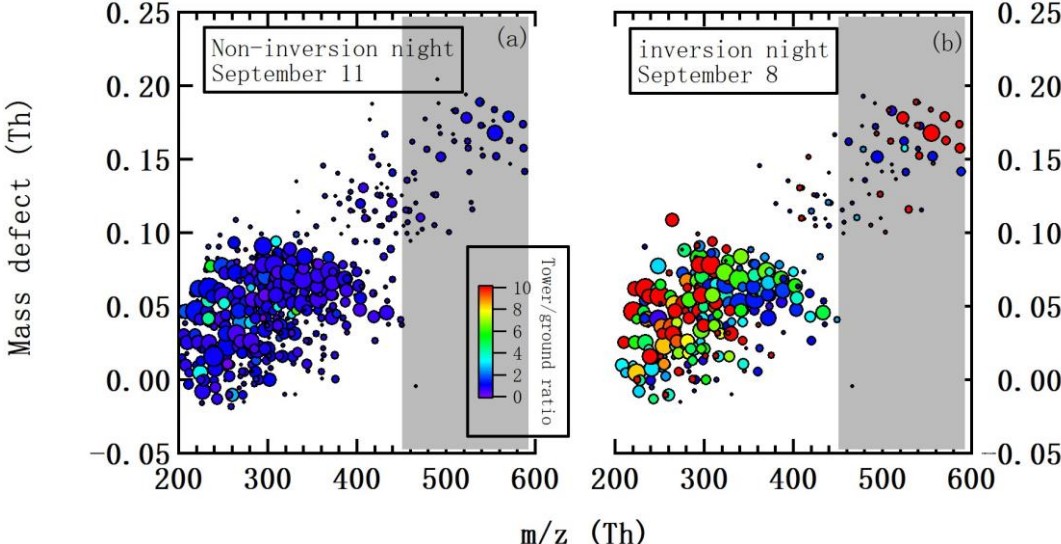

Figure 7. Mass defect (MD) plots of the selected (a) "non-inversion night" case (September 11), and (b) "inversion night" case (September 8). Color code indicates the ratios between tower/ground HOM concentrations. Grey shade area denotes the dimer range (*m/z* 450-600).

Examination of the selected HOM molecules was useful and efficient to assess the changes in HOMs, however, such an analysis might only indicate the major formation pathways. Hence, it was also worthwhile to have a holistic view of the entire mass spectra and all the detected HOMs. The mass defect (MD) plot (Figure 7) separates all identified compounds according to their exact masses on the x-axis and the deviation from the integer mass on the y-axis. Each circle represents a compound, with the areas scaled by concentrations, and colored by the ratios between tower and ground concentrations. Figure 7a and 7b are MD plots showing the mean spectra of the selected non-inversion night (September 11) and inversion night (September 8), respectively. Without the formation of a decoupled layer, nearly the same concentration distributions of HOMs were observed. In contrast, during the inversion night (September 8, Figure7b), large differences could be found between the two measurement heights. Moreover, a significant fraction of the ground HOMs disappeared on the inversion night, and the concentrations of the remaining HOMs were also lower, confirming the aforementioned results obtained with the selected HOM groups.

## 4.4 Study limitations

Several limitations still exist in this study. From the measurement side, one major concern was the comparability between our two CI-APi-TOF mass spectrometers. In the worst case, our conclusion might be biased if instrument responses changed due to some parameter that correlated with the observed inversions. The main parameters in this case would be ambient temperature and RH. As both instruments were located in temperature-controlled containers and the sample flow was mixed 1:2 with dry sheath air in the CI-APi-TOF drift tube, neither of these were expected to yield such large changes. However, for confirmation, we compared the detailed spectral evolution during days and nights of the study. Figure 8 shows an example of hourly changes of the ratios between tower and ground HOMs, over a 24h period without nighttime temperature inversion (September 11). During this period, ambient temperatures changed from 19.1 °C (12:00 LT) to 8.8 °C (07:00 LT) at ground level, and from 17.9 °C to 8.1 °C at tower level. Ambient RHs also increased from 72 % to 96 % at ground level, and from 74 % to 98 % at tower level. While some scatter is visible in the 200-300 Th range during some parts of the night, good agreement was observed between the two instruments throughout the night, despite large variability in temperatures and RHs.

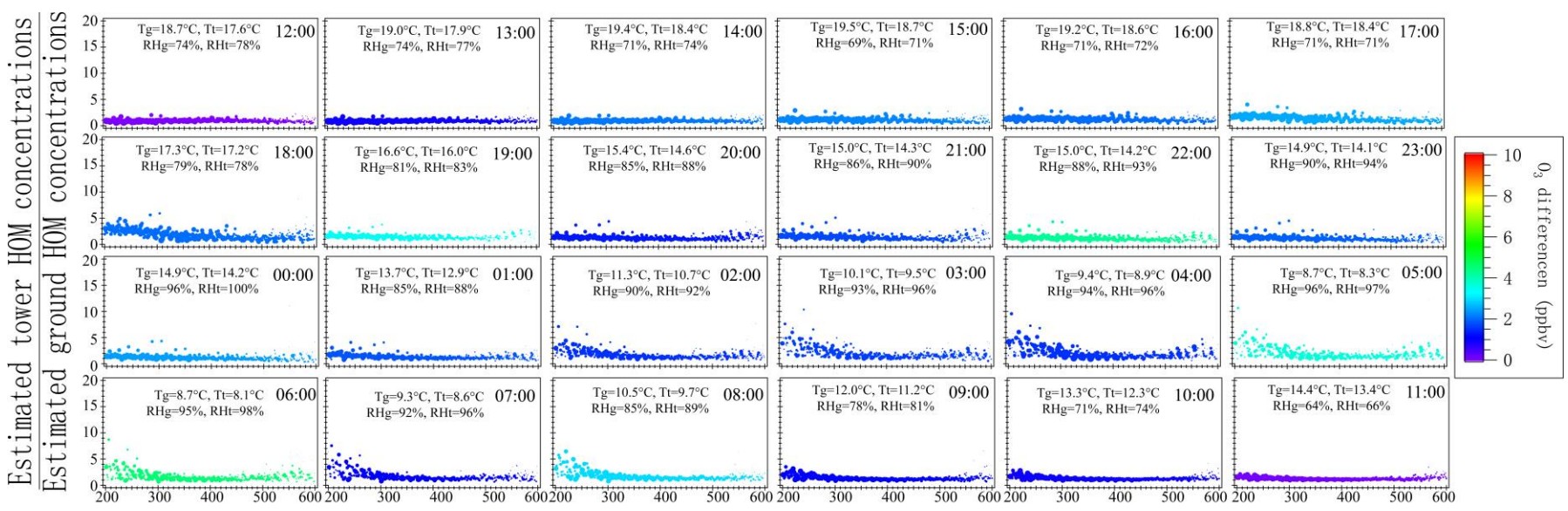

Figure 8. Hourly changes of the ratios between estimated tower and ground HOM concentrations from September 11, 12:00 to September 12, 11:00 (non-inversion night). Markers are sized by ground HOM concentrations and colored by $O_3$ difference between tower and ground $(O_{3_{tower}} - O_{3_{ground}})$. Hourly ambient temperatures at ground (Tg) and tower (Tt) levels, and RH at ground (RHg) and tower (RHt) levels are shown in each subplot.

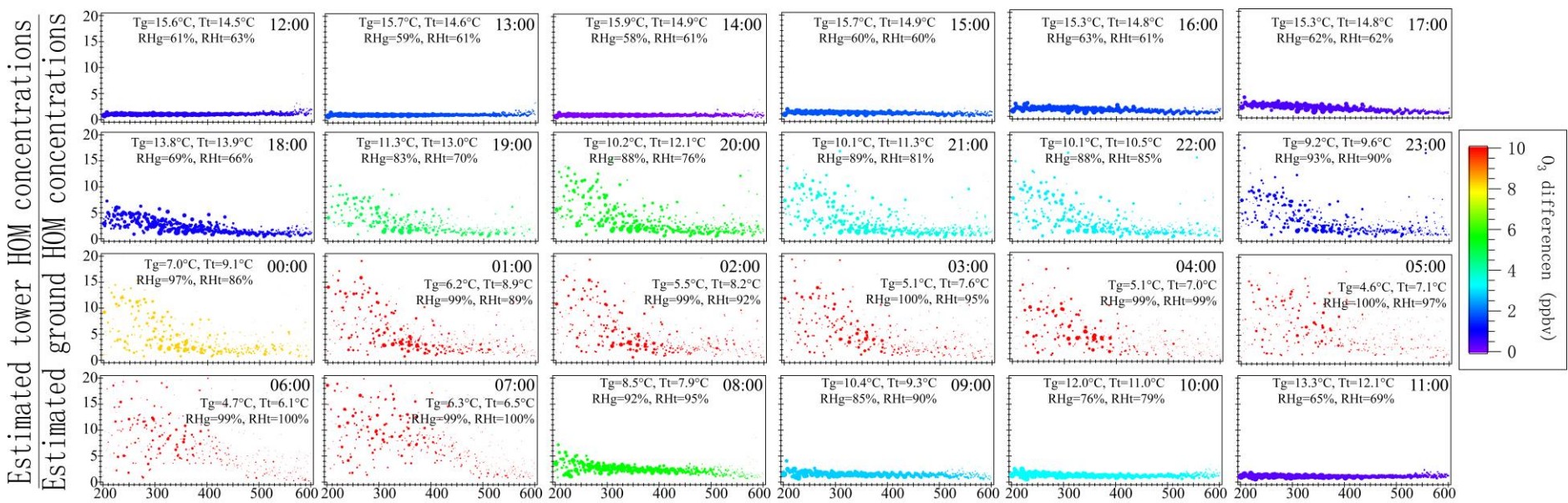

Figure 9. Hourly changes of the ratios between estimated tower and ground HOM concentrations from September 8, 12:00 to September 9, 11:00 (inversion night). Markers are sized by ground HOM concentrations and colored by $O_3$ difference between tower and ground ($O_{3_{tower}} - O_{3_{ground}}$). Hourly ambient temperatures at ground (Tg) and tower (Tt) levels, and RH at ground (RHg) and tower (RHt) levels are shown in each subplot.

In contrast, during a 24h period with nighttime temperature inversion (September 8,
shown in Figure 9), the ratios agreed well only during daytime (from 12:00 to 17:00,
and 09:00-11:00 on the next day). Between these two periods, temperature and RH were
most of the time in the same range as on September 11 (when no strong deviations were
observed), but now the HOM behavior changed dramatically between the two heights.
The ratios increased from ~1 (during daytime) up to ~20 at 07:00 for some of the
measured molecules.

Figures 8 and 9 clearly imply that the large differences between ground and tower HOM
concentrations were driven by temperature inversions and consequent changes in the
composition of the air in the two detached layers. Large changes in HOMs were
observed only when the ground temperature was lower than the tower temperature and
when the ozone concentration at ground level was several ppb lower than in the tower
(shown as color scale in Figure 8 and 9). Absolute temperatures or RHs at the two
heights were not able to explain the changes. As a concrete example, good agreement
was observed at 07:00, September 12, while ambient temperatures were low (ground
and tower temperatures were 9.3 °C and 8.6 °C, respectively) and RHs were high
(ground and tower RHs were 92 % and 96 %, respectively), but large deviations were
found at 20:00, September 8, when higher temperatures (ground and tower temperatures
were 10.2 °C and 12.1 °C, respectively) and lower RHs (ground and tower RHs were
88 % and 76 %, respectively) were observed. In other words, neither low temperatures
nor high RHs caused large changes to our instruments. Instead, the large discrepancies
between the two CI-APi-TOFs were only observed when other key parameters (like
ozone) were found to deviate considerably between the two heights.

From the micrometeorology side, the contribution from the potential
micrometeorological processes in the layer between 1.5 m and 4.2 m (between the
sampling heights of the ground HOMs and other parameters) could not be estimated
with the current experiment design (i.e., only two measurement heights). Similarly, the
influence from horizontal advection could not be entirely ruled out as a reason for the
reduced ground-level HOM concentrations (and other significantly changed species),
because of the possible horizontal inhomogeneity of HOM precursors and oxidants
below the canopy. However, our conclusion was confirmed by the incompatibility
between the increasing ground level MT and $CO_2$ concentrations and the advection
hypothesis (i.e., all species would show similar tendencies if advection played a major
role), indicating the influence of horizontal and vertical advection is probably minor
when compared to the increasing sink. However, conclusive evidence is still needed
which highlights the need for joint vertical-planar HOM studies, measuring both
vertical and horizontal distribution of HOM concentrations.

**5   Conclusion**
Highly oxygenated molecules (HOMs) were measured above the canopy and at ground
level (below the canopy) in a boreal forest environment during the IBAIRN campaign
that took place in September 2016. Boundary layer dynamics and micrometeorology
were found to be important factors that influence the abundance and trends of HOMs
at ground level, by perturbing both their sources and sinks. In the well-mixed boundary
layer (e.g. during daytime or nights without strong inversion), HOM concentrations and
other measured species were overall similar between the ground and tower
measurements. In contrast, much lower ground level HOM concentrations were
observed when nighttime temperature inversion and formation of a decoupled layer
occurred below the canopy. On one hand, the production of the ground-level HOMs
could be affected by the decreasing $O_3$ concentrations and the increasing MT
concentration in the shallow layer. On the other hand, the surface area to volume ratio
dramatically increased in the shallow layer compared to the nocturnal boundary layer.
The possibility of losses on surfaces for ground-level HOMs became much larger than
usual during inversion nights. The enhanced interaction of air in the decoupled layer
with the forest floor was supported by increased concentrations of $CO_2$, emitted mainly
from the ground, in this layer.

We have presented the first detailed measurements of HOMs below and above the
canopy across a wide range of atmospheric stability conditions. The results highlight
the significance of near-ground boundary layer dynamics and micrometeorological
processes to the ambient HOMs, showing that ground-based HOM measurement at this
site might not be representative for the entire nocturnal boundary layer. Conventionally,
field measurements of HOMs and other parameters are mostly performed close to the
ground, and the possible effect of boundary layer dynamics and micrometeorological
processes to the HOM concentrations have rarely been considered. Aerosol particle
growth and SOA formation rates at ground level are likely to be influenced by the
reduced HOM concentrations in the inversion nights. However, there are still
limitations due to current experiment design, such as horizontal separation in
instrument set-up, or the uncertainties from using point measurements at two heights to
infer larger scale exchange. Clearly, more vertical and planar measurements of HOMs
are needed to confirm the emerging picture presented here. Influence of boundary layer
dynamics should be better characterized and evaluated in future field campaigns.

**Acknowledgements**
This work was supported by the IBAIRN project, the Academy of Finland Center of
Excellence in Atmospheric Science, European commission Actris2 and Actris PPP, the
European Research Council (Grant 638703-COALA), transnational access from
ENVRI plus, and SMEAR II technical team. Q.Z. thanks ATM-DP (Doctoral Program
in Atmospheric Sciences) graduate programs, John Crowley and Max Plank Institute in
association with IBAIRN proposal, and the tofTools team for providing tools for mass
spectrometry analysis. G.K. acknowledges the support from the U.S. National Science
Foundation (NSF-EAR-1344703, NSF-AGS-1644382), the U.S. Department of Energy
(DE-SC0011461), and University of Helsinki for supporting a 3-month sabbatical leave
at the Division of Atmospheric Sciences.

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
