# Peer review of "Vertical characterization of Highly Oxygenated Molecules (HOMs) below and above a boreal forest canopy 2 3"

_Atmospheric Chemistry and Physics, 2017_

## Referee Comment (RC1) · Anonymous Referee #1 · 13 Mar 2018

Review of "Vertical Characterization of Highly Oxygenated Molecules (HOMs) below and above a boreal Forest Canopy" by Zha et al.

This manuscript describes analyses of data collected on the abundance of highly oxidized molecules in a forested environment as part of a comprehensive field campaign. Key to the analysis are observations above and below the forest canopy, which, in principle, allows assessment of flux terms applicable to HOMs. The approaches are well-described and appropriate caveats are clearly stated.

The main issue of concern to this reviewer is the use of two instruments for measurement of HOMs (CP-APi-TOF instruments) that were not side-by-side intercompared over the range of molecular and radical species in the laboratory nor in the field. A single sentence (lines 179-181) states that a laboratory intercomparison was conducted with a permeation tube (not stating what molecules were emitted by the tube) and the results showed good agreement with the relative transmission efficiency experiments. In this approach, assumptions were made as to when the two instruments at two heights (1 m and 36 m) should agree. From this, relatively sensitivities between the two instruments were derived (pages 7 and 8).  The sensitivity ratio of the two instruments ranged from about 2 to more than 10, depending on the m/z values. Such large differences require significantly more experiments and demonstration than presented in this paper, to convince the reader that the conclusions that arise are valid. This reviewer sees this as a fatal flaw in this manuscript. This is a major point that leads to the recommendation of reconsideration of this manuscript after revisions to address this important point. There are also other issues that should be addressed, as described below.

Page 2, line 32.  Suggest rewording "…attached to the forest floor."
Page 2, line 37.  Suggest "This could, in turn, influence interpretation of the growth…"
Page 3, line 60.  Were the HOM clusters in the laboratory experiments also "naturally charged"?
Page 3, line 79.  Suggest "…compounds, with masses between…"
Page 3, line 84.  Suggest "…compounds, with masses between…".  In the dimerization of $RO_2$ radicals, what is the chemical mechanism, and what types of molecules are formed (peroxy radicals, organic peroxides, ROOR, etc.)?
Page 4, line 91.  Suggest rewording.  Do you mean oxidants of monoterpenes that product HOMs, or oxidants of HOMs producing other molecules or radicals?
Page 4, line 95.  Is the term "sub-canopy" typical used, or would "below canopy" and/or "in canopy" be better?
Page 4, line 108.  It is stated that the lower inlet is at 1.5 m, which is different than stated on page 5, line 131.  Suggest making everything consistent.
Page 5, line 118-119.  Suggest "…southeast of the site, and from the city area of Tampere…"
Page 5, line 124.  Should "April" be "August"?
Page 5, line 128-129.  Suggest "…deployed at the top…"
Page 6, line 138.  Suggest rewording "…then converged to the center…"
Page 6, line 146.  Suggest "..stack of ion lenses guided the ions…"
Page 6, line 153.  Are you missing a summation sign before the "M" in the numerator?
Page 6, line 164-165.  While it is true that absolute HOM concentrations are not as important in this work, the relative sensitivity of the two instruments is of critical importance (see earlier comment).
Page 7, line 180-181.  Since the permeation tube experiments could potentially be very important, much more detail needs to be given.  What is(are) the compound(s) coming from the permeation tube?  What does "good" agreement mean?  Can the results be included in Figure 1?
Page 7, line 184.  How is the instrument tuned for maximum sensitivity at the largest masses?

Page 8, Figure 1. Is there a theoretical reason to fit the data with a power law, or did that simply provide a reasonable representation?

Page 8, lines 195 to 197. It is concerning that the various controlling parameters were not measurements at precisely the same heights as the HOM measurements. It is also concerning that these other measurements were 100 m away. Can you provide information that these differences did not affect the conclusions of this study?

Page 8, line 200. Suggest "…with a lower detection limit…"

Page 8, line 202. Suggest "…that had a lower detection limit…"

Page 8, line 205. Suggest"…lower detection limit of the NO$_x$ analyzer was…"

Page 9, line 209. Also concerning that the aerosol measurements were not made at the same heights at the HOM measurements. What impact could this have?

Page 9, line 216. Suggest "…averaging intervals, except for the MT (in 1-hour averaging intervals)."

Page 9, line 227-229. Suggest "The mean air temperature and RH observed at ground level were…, and at the tower level were…".

Page 9, line 230-231. Suggest providing statistics for temperature, RH, and O$_3$ separately for daytime and nighttime.

Page 9, line 222 and page 10, line 223. This statement is confusing. The NO$_x$ detection limit is 50 pptv (line 205), so how does this relate to mean +/- standard deviation values given. Need a bit more text to describe what was done statistically, and what the results say.

Page 20, line 235-236. Suggest "…were generally higher than those above…"

Page 11, line 242. Since the transmission efficiencies are not used in the reduction of the data, this reviewer disagrees with the statement that the sum of the signals between m/z 200 to 600 represents the total HOM concentration. This needs some reworking. If the transmission efficiencies are not know, then suggest not giving HOM concentrations, but perhaps HOM signals.

Page 11, line 246-248. Are the statistics for all the data, or just daytime or nighttime? The value after +/- is presumably the standard deviation. This needs to be stated. In the last sentence "these differences" are mentioned, but it needs to be specifically stated which differences are being referred to (e.g. differences in the means above and below (fairly small), differences in the medians (larger), etc.)?

Page 11, line 252. Suggest adding statistics to demonstrate HOM concentrations at the two heights were not different during the day.

Page 11, line 254. Suggest a figure showing that the ratio (or some other metric) of the HOM concentrations at the two heights did not change with time during the day.

Page 11, line 257. Suggest including statistics and time dependence for the two heights (as above) for nighttime data.

Page 11, line 258. Suggest showing the temperature difference between the two heights in Figure 2 to help clearly show when there are temperature inversions.

Page 11, line 261 and Figure 3. Suggest giving statistics to support the statement that there is good agreement around midday.

Page 11, line 264 and Figure 3. Suggest giving statistics for nighttime HOM concentrations to support the statement.

Page 12, line 270. Suggest "…shows the mean mass spectra…" and "…UMR, for m/z 200…" and remove "HOM measurements"

Page 12, line 283. Suggest "…strength and/or source-sink…"

Page 14, line 292. Suggest adding "likely" or "probably" in "…level are likely influenced by….".

Page 14, line 294-296. Suggest "…the potential impact of such micrometeorological phenomena on ground level HOMs, for the nights during the campaign without precipitation or instrument failure, were selected…"

Page 14, line 297. Suggest "…based on the occurrence of temperature inversions…"

Page 14, line 299.  Suggest "…type category consisted of 6 nights…".
Page 14, line 301.  Suggest "…lower than tower…"
Page 14, line 307.  Suggest "…above the canopy was relatively…"
Page 14, line 310.  Is it known that there are higher VOC emissions near the ground within forest canopies?  A reference or two would be good here.
Page 14, line 315.  Suggest "…similar in both categories and heights…"
Page 16, Table 1.  This reviewer found the gray bars in the table (not the titles) confusing.  Suggest configuring the table differently.
Page 17, line 344 and Table 2.  Were these categories done for all conditions, all times (looks like it is nighttime), and both heights?
Page 17, line 363.  Suggest "Roughly, Ri values in excess…" and "…stratified conditions appreciably…".
Page 17, last paragraph and Figure 5.  Were these data for the ground level measurements?
Page 19, line 381.  Suggest "…significant decreases after midnight."
Page 19, line 387.  Suggest "…HOM concentrations might…"
Page 19, line 388.  Suggest "…but also due to some other processes such as additional losses."
Page 19, line 391.  Suggest giving the location of the Alekseychik et al study.
Page 19, line 392.  Suggest "…Ri conditions in the…"
Page 19, line 396.  Suggest "…in significantly different $O_3$…"
Page 19, line 397.  See earlier comment about "sub-canopy".
Page 20, top paragraph.  This reviewer found the use of $T_1$, $T_2$, etc confusing since capital T is usually reserved for temperature.  Suggest using different symbols.
Page 20, line 418.  Suggest "Note that these…"
Page 20, line 421.  Suggest "area-to-volume ratio…".
Page 20, line 424.  See earlier comment about "sub-canopy".
Page 22, line 452.  Suggest "…however, such an analysis might only indicate the major…"
Page 22, line 453.  Suggest "…holistic view of the entire mass spectrum…" or some other rewording.
Page 23, line 461.  Suggest "…large differences could…"
Page 23, line 463.  Suggest "…disappeared on the…"
Page 23, line 468.  Suggest "…limitations still exist in this…"
Page 23, line 472.  Suggest "…influence of horizontal advection could not be entirely ruled out as a contributor to…"
Page 23, line 473.  Suggest "…HOM concentrations…".  Suggest rewording "largely changed species"
Page 23, line 473.  Suggest "…because of possible horizontal…"
Page 23, line 477.  Suggest "…advection is probably minor…"
Page 23, line 479.  Suggest "…evidence is still needed…"
Page 23, line 479-280.  Suggest "…which highlights the need for…".  Also define "joint vertical-planar HOM studies".
Page 23, line 484.  Suggest "…IBAIRN campaign that took place in September 2016."
Page 24, line 486.  Suggest "…that influence the abundance and trends of HOMs…"
Page 24, lines 505-507.  Suggest "…close to the ground, and the effect of boundary layer…" and "…processes to HOM concentrations have…"
Page 24, line 509.  Suggest "…HOM concentrations found in nocturnal inversion situations."
Page 24, line 510.  Suggest "Influence of boundary…"
Page 27, line 582.  There is a typo in this reference.

---

## Referee Comment (RC2) · Anonymous Referee #2 · 25 Mar 2018

The article by Zha et al. touches on important but challenging issues of how the extremely low volatility, highly oxidized molecules (HOMs aka ELVOC) behave below and above the tree canopy. The article is generally clear and makes many confident insights into the atmospheric variability of these molecules. Because these molecules have high propensity to form SOA, such measurements are needed and potentially valuable. However, I would have some important reservations to this version of the manuscript about the data interpretation and QC/QA, but hopefully, this can be successfully addressed by the authors.

[Figure]

*Major*

1) Two heights were chosen for measurements of HOMs by two different CI-APi-TOF instruments. Although I agree with the other reviewer that this might be regarded as a flaw, I think it does not have to be a critical flaw as long as there is a substantial effort to ensure that the two instruments were in perfect agreement. I am surprised why a portion of measurements was not conducted at the same height (either ground or tower) by both instruments side by side first before moving on to measuring at two heights. It is unfortunate because the collocated measurements would help to ensure that the data reported by two instruments are indeed identical. I am concerned that it might be difficult otherwise to demonstrate this, because there are numerous factors that may affect the agreement between two different instruments apart from the relative transmission. For example, fragmentation, clustering, declustering, or other processes modulated by changes in ambient temperature and humidity may have differed as a function of day, as a function of height, and over longer time scales in either or both instruments. It is therefore really difficult to get convinced if the differences are necessarily because of the height and not because of the differences in each instrument's quantification. A single point laboratory comparison of just the relative transmission of two instruments does not seem sufficient, because of the inability of assessing the factors which change over time. It would be ideal to calibrate regularly the instruments independently and assess separately the deviation from the theoretical transmission of the TOFs like in Heinritzi et al. (2016). Then you could compare the datasets and see how consistent they would be in the middle of a day and at night.

2) Eq. 1 is only valid if the conditions in the reaction chamber or TOF chamber have not changed. Small variations in pressure, temperature or humidity could affect the calibration coefficient. It would be inappropriate to expect that if you set the collision rate the same for all the ions, the relative consistency in two instruments will be the same, because the sensitivity can change over time also because of the drift parameters and resulting issues such as different fragmentation which would not be accounted for by the single transmission correction. It is therefore recommended to take zeros

and calibrations of the instruments frequently and none of these are shown except that permeation device is mentioned.

3) I understand that the authors did not seem concerned about the absolute quantification (L165-166) but even the relative quantification is uncertain if one of the instrument was affected by different conditions or its sensitivity drifted throughout a day. It is unclear how the instruments were housed and if the temperature inside the instruments were monitored and if it was consistent at the two heights. A visual schematic would be useful. One way around could be to either add a period of collocated measurements for a few days and compare the range of compounds or show how stable baselines, sensitivities, and transmissions, were throughout the measurement period if you did regular calibrations.

4) Why do you assume a noon period should have identical HOMs concentrations at two heights? I thought these were the first measurements of HOMs at two heights or can you provide a reference? Again, I am surprised that this period was chosen for transmission comparison instead of collocating the two instruments for a longer period. I am not convinced that it is fair to assume that all the HOMs will have no concentration gradient during a day. I am worried that if the HOMs are lost more rapidly at the surface the relative transmission ratio between the instruments may be biased towards larger masses which could be expected to be lost as a function of mass-dependent volatility which could mimic the duty-cycle and transmission related mass discrimination.

5) Fig. 2 shows a good agreement for HOMs during noon and not so good otherwise. This relative scaling seems rather arbitrary and not completely unsurprising given the data were normalized using Figure 1 which was derived for the noon time. I am skeptical if these differences really represent the effect of different heights. The differences are huge because the data are shown on a log scale. I would recommend showing the comparison on a linear scale.

6) Overall, I was missing a stronger link to chemical and physical properties of HOMs

and a deeper insight into individual classes and not just the total sum of HOMs. In particular, the transport could be molecule specific and may not be unified across the full range of HOM vapor pressures. I feel that the dataset has a much higher potential for teaching a reader about the behavior of HOMs. The current version of the paper gives an impression of semiquantitative and speculative in terms of HOM vertical behavior. I am not convinced by looking at temperature variations (Fig. 2) that the inversion hypothesis is strong. There is much attention directed to the general remarks about advection and turbulent quantities which is difficult to infer how they affect HOMs without direct flux measurements of HOMs by eddy covariance.

*Minor*

7) I wonder why you are adopting the HOM nomenclature and not ELVOC. Mentel et al. (2015) suggested that the latter is more appropriate when referring to atmospheric impacts and HOMs if the focus is more on compounds' structure. As you are focusing on the behavior in the atmosphere I think considering volatilities could make sense. Are you sure that all the molecules you report are highly oxidized (high carbon oxidation state)? That would not be true for compounds such as cyclic and linear siloxanes which should be subtracted from the HOMs class.

8) Figure 3 does not provide much information when it is colored by the time of day. For the reasons mentioned above the scatter outside of the noon hour might not necessarily be because of less good mixing other than at noon. One could possibly learn more if the data were colored by potential temperature or relative humidity.

9) Figure 4c, the scatter looks weird that it is so much shifted (by an order of magnitude) but still reasonably correlated. I wonder if it is possible to evaluate any volatility dependent difference but it seems that it might be difficult if there is a high uncertainty in transmission differences.

10) Figure 1 is technical and could be moved to SI. I would suggest to replace it

by separate individual curves of theoretical and measured transmissions from each instrument and the datasets should be corrected individually.

**Technical**
L67 space between ")" and "from".

Table 1, ensure the number of significant figures is consistent and as appropriate.

Table 2, if there is only one N atom in a molecule there is no need to add 1. For example, $C_{10}H_{15}O_{11}N_1$ should be $C_{10}H_{15}O_{11}N$.

**References**

Heinritzi, M., Simon, M., Steiner, G., Wagner, A. C., Kürten, A., Hansel, A., and Curtius, J.: Characterization of the mass-dependent transmission efficiency of a CIMS, Atmos. Meas. Tech., 9, 1449-1460, https://doi.org/10.5194/amt-9-1449-2016, 2016.

Mentel, T. F., Springer, M., Ehn, M., Kleist, E., Pullinen, I., Kurtén, T., Rissanen, M., Wahner, A., and Wildt, J.: Formation of highly oxidized multifunctional compounds: autoxidation of peroxy radicals formed in the ozonolysis of alkenes – deduced from structure–product relationships, Atmos. Chem. Phys., 15, 6745-6765, https://doi.org/10.5194/acp-15-6745-2015, 2015.

---

## Author Comment (AC1) · 12 May 2018

*We thank both reviewers for their valuable comments. The reviewers' comments are listed below and are followed by our replies (in italics).*

Anonymous Referee #1

This manuscript describes analyses of data collected on the abundance of highly oxidized molecules in a forested environment as part of a comprehensive field campaign. Key to the analysis are observations above and below the forest canopy, which, in principle, allows assessment of flux terms applicable to HOMs. The approaches are well-described and appropriate caveats are clearly stated.

The main issue of concern to this reviewer is the use of two instruments for measurement of HOMs (CP-APi-TOF instruments) that were not side-by-side intercompared over the range of molecular and radical species in the laboratory nor in the field. A single sentence (lines 179-181) states that a laboratory intercomparison was conducted with a permeation tube (not stating what molecules were emitted by the tube) and the results showed good agreement with the relative transmission efficiency experiments. In this approach, assumptions were made as to when the two instruments at two heights (1 m and 36 m) should agree. From this, relatively sensitivities between the two instruments were derived (pages 7 and 8). The sensitivity ratio of the two instruments ranged from about 2 to more than 10, depending on the m/z values. Such large differences require significantly more experiments and demonstration than presented in this paper, to convince the reader that the conclusions that arise are valid. This reviewer sees this as a fatal flaw in this manuscript. This is a major point that leads to the recommendation of reconsideration of this manuscript after revisions to address this important point. There are also other issues that should be addressed, as described below.

*The reviewer is indeed correct to point out this concern. We shared this concern when starting our initial analyses and critically examined the mass spectra from both instruments with this in mind already before proceeding with the analyses presented in the original manuscript. In the process we became convinced that the differences were indeed real, and not purely instrumental artifacts. Unfortunately, we failed to include enough of such data in the manuscript, as both reviewers pointed out. This has now been amended in the revised version, and we believe that we now clearly show that the differences are due to variations in atmospheric composition between the two heights, and not a result of instrumental changes.*

*The instrument in the tower had been part of the site's continuous measurements since 2014, and the ground instrument was deployed in 2016. It is extremely unfortunate that a direct side-by-side inter-comparison was not done at the time of the deployment, as it would have greatly facilitated the data analysis. However, as a first*

*step to validate a comparison, we used a permeation tube with trinitrotriazinane, detected as $C_3H_6N_6O_6 \cdot NO_3^-$ (m/z 284) and $(C_3H_6N_6O_6)_2 \cdot NO_3^-$ (m/z 506) located in the HOM monomer and dimer range, respectively, and the same permeation source was connected in the same manner first to one instrument and then to the other. The results supported the increasing relative transmission efficiency (TE) curve presented in Fig. 1.*

*Now we added these inter-comparison results to Figure 1 and modified the text to:*

*"Additionally, an inter-comparison between the two instruments with a permeation tube containing trinitrotriazinane ($C_3H_6N_6O_6$) was conducted in the field right after the campaign. The results showed good agreements with the relative TE, lending confidence to the method used here…"*

*To validate the large differences (nearly an order of magnitude) between observed HOM concentrations during inversion nights, we also added two figures and text in section 4.4 to show more temporal and spectral details of the changes between the instruments. We believe the presented figures unambiguously show that the changes are a result of different chemistry at the two heights due to decoupled layers, rather than e.g. temperature or RH-driven instrumental changes (as suggested by reviewer 2). The added section is inserted below:*

*"From the measurement side, one major concern was the comparability between our two CI-APi-TOF mass spectrometers. In the worst case, our conclusion might be biased if instrument responses changed due to some parameter that correlated with the observed inversions. The main parameters in this case would be ambient temperature and RH. As both instruments were located in temperature-controlled containers and the sample flow was mixed 1:2 with dry sheath air in the CI-APi-TOF drift tube, neither of these were expected to yield such large changes. However, for confirmation, we compared the detailed spectral evolution during days and nights of the study. Figure 8 shows an example of hourly changes of the ratios between tower and ground HOMs, over a 24h period without nighttime temperature inversion (September 11). During this period, ambient temperatures changed from 19.1 ℃ (12:00 LT) to 8.8 ℃ (07:00 LT) at ground level, and from 17.9 ℃ to 8.1 ℃ at tower level. Ambient RHs also increased from 72 % to 96 % at ground level, and from 74 % to 98 % at tower level. While some scatter is visible in the 200-300 Th range during some parts of the night, good agreement was observed between the two instruments throughout the night, despite large variability in temperatures and RHs.*

[Figure]

*Figure 8 Hourly changes of the ratios between estimated tower and ground HOM concentrations from September 11, 12:00 to September 12, 11:00 (non-inversion night). Markers are sized by ground HOM concentrations and colored by O₃ difference between tower and ground ﹙$O_{3_{tower}} - O_{3_{ground}}$﹚. Hourly ambient temperatures at ground (Tg) and tower (Tt) levels, and RH at ground (RHg) and tower (RHt) levels are shown in each subplot.*

[Figure]

*Figure 9 Hourly changes of the ratios between estimated tower and ground HOM concentrations from September 8, 12:00 to September 9, 11:00 (inversion night). Markers are sized by ground HOM concentrations and colored by $O_3$ difference between tower and ground ($O_{3_{tower}} - O_{3_{ground}}$). Hourly ambient temperatures at ground (Tg) and tower (Tt) levels, and RH at ground (RHg) and tower (RHt) levels are shown in each subplot.*

*In contrast, during a 24h period with nighttime temperature inversion (September 8, shown in Figure 9), the ratios agreed well only during daytime (from 12:00 to 17:00, and 09:00-11:00 on the next day). Between these periods, temperature and RH were most of the time in the same range as on September 11 (when no strong deviations were observed), but now the HOM behavior changed dramatically between the two heights. The ratios increased from ~1 (during daytime) up to ~20 at 07:00 for some of the measured molecules.*

*Figures 8 and 9 clearly imply that the large differences between ground and tower HOM concentrations were driven by temperature inversions and consequent changes in the composition of the air in the two detached layers. Large changes in HOMs were observed only when the ground temperature was lower than the tower temperature and when the ozone concentration at ground level was several ppb lower. Absolute temperatures or RHs at the two heights were not able to explain the changes. As a concrete example, good agreement was observed at 07:00, September 12, while ambient temperatures were low (ground and tower temperatures were 9.3 ℃ and 8.6 ℃, respectively) and RHs were high (ground and tower RHs were 92 % and 96 %, respectively), but large deviations were found at 20:00, September 8, when higher temperatures (ground and tower temperatures were 10.2 ℃ and 12.1 ℃, respectively) and lower RHs (ground and tower RHs were 88 % and 76 %, respectively) were observed. In other words, neither low temperatures nor high RHs caused large changes to our instruments. Instead, the large discrepancies between the two CI-APi-TOFs were only observed when other key parameters (like ozone) were found to deviate considerably between the two heights."*

Page 2, line 32.   Suggest rewording "…attached to the forest floor."
*Modified.*

Page 2, line 37.   Suggest "This could, in turn, influence interpretation of the growth…"
*Modified.*

Page 3, line 60.   Were the HOM clusters in the laboratory experiments also "naturally charged"?
*Yes.*

Page 3, line 79.   Suggest "…compounds, with masses between…"
*Modified.*

Page 3, line 84.  Suggest "…compounds, with masses between…".  In the dimerization of RO2 radicals, what is the chemical mechanism, and what types of molecules are formed (peroxy radicals, organic peroxides, ROOR, etc.)?

*Modified.*
*The exact chemical mechanism of $RO_2$ radical's dimerization is still under discussion, and therefore we did not go into details about this in the manuscript. However, recent studies (Ehn et al., 2014; Jokinen et al., 2014; Berndt et al., 2018) have shown strong support for the following pathway:*

$$RO_2 + R'O_2 \rightarrow ROOR' + O_2$$

*which is the "HOM dimer" in the manuscript. We also add two more references (Jokinen et al., 2014; Berndt et al., 2018) to the sentence now.*

Page 4, line 91.  Suggest rewording.  Do you mean oxidants of monoterpenes that product HOMs, or oxidants of HOMs producing other molecules or radicals?
*Modified. Now the sentence is*

*"Unsurprisingly, the oxidants producing HOMs (e.g. $O_3$) were found almost uniformly distributed within the well-mixed daytime boundary layer".*

Page 4, line 95.  Is the term "sub-canopy" typical used, or would "below canopy" and/or "in canopy" be better?
*Modified. Now the three terms are unified as "below the canopy".*

Page 4, line 108.  It is stated that the lower inlet is at 1.5 m, which is different than stated on page 5, line 131.  Suggest making everything consistent.
*Modified. The height of lower inlet is ~1.5m.*

Page 5, line 118-119.  Suggest "…southeast of the site, and from the city area of Tampere..."
*Modified.*

Page 5, line 124.  Should "April" be "August"?
*According to the cited references, "April" is correct because nocturnal boundary layer has not been measured in August.*

Page 5, line 128-129.  Suggest "…deployed at the top…"
*Modified.*

Page 6, line 138.  Suggest rewording "…then converged to the center…"
*Modified. Now the expression is "then centered to an ion reaction tube".*

Page 6, line 146.  Suggest "..stack of ion lenses guided the ions…"
*Modified.*

Page 6, line 153. Are you missing a summation sign before the "M" in the numerator?

*Corrected.*

Page 6, line 164-165. While it is true that absolute HOM concentrations are not as important in this work, the relative sensitivity of the two instruments is of critical importance (see earlier comment).

*See our first response above.*

Page 7, line 180-181. Since the permeation tube experiments could potentially be very important, much more detail needs to be given. What is(are) the compound(s) coming from the permeation tube? What does "good" agreement mean? Can the results be included in Figure 1?

*See our first response above.*

Page 7, line 184. How is the instrument tuned for maximum sensitivity at the largest masses?

*The sensitivity of a CI-APi-TOF can be tuned to maximize ion throughput at different mass ranges by varying voltages and radiofrequencies applied to the guiding quadrupoles in the instrument. Especially the quadrupole settings can increase the throughput at larger masses at the expense of the smaller masses. In this study, the tower CI-APi-TOF was tuned for maximum sensitivity at the highest masses.*
*We assumed a detailed discussion on this would become too technical in the manuscript, but we now reformulated the sentence to:*

*"...the tower CI-APi-TOF had been tuned for higher sensitivity at the larger masses".*

Page 8, Figure 1. Is there a theoretical reason to fit the data with a power law, or did that simply provide a reasonable representation?

*No, it just provided a reasonable representation.*

Page 8, lines 195 to 197. It is concerning that the various controlling parameters were not measurements at precisely the same heights as the HOM measurements. It is also concerning that these other measurements were 100 m away. Can you provide information that these differences did not affect the conclusions of this study?

*Thousands of different parameters are measured at the SMEAR II station, and therefore it is inevitable that not all of them are co-located. We could not totally rule out the influence from other micrometeorological processes occurring in the space between the different locations/heights, however, their contributions should be minor compared to the dramatic changes observed in HOM concentrations. This matter was also discussed in the study limitation part (section 4.4):*

*"…the contribution from the potential micrometeorological processes in the layer between 1.5 m and 4.2 m (between the sampling heights of the ground HOMs and other parameters) could not be estimated with the current experiment design (i.e., only two measurement heights). Similarly, the influence from horizontal advection could not be entirely ruled out as a contributor to the reduced ground-level HOM concentrations (and other significantly changed species), because of the possible horizontal inhomogeneity of HOM precursors and oxidants below the canopy. However, our conclusion was confirmed by the incompatibility between the increasing ground MT and $CO_2$ concentrations and the advection hypothesis (i.e., all species would show similar tendencies if advection played a major role), indicating the influence of horizontal and vertical advection is probably minor when compared to the increasing sink."*

*It could also be added that the main conclusion of our manuscript is that fixed-point ground level observations should not automatically be assumed to always represent the situation at higher altitudes. If large discrepancies were taking place also when moving ~100m in the horizontal direction, this would only make our conclusions even more important.*

Page 8, line 200.    Suggest "…with a lower detection limit…"
*Modified.*

Page 8, line 202.    Suggest "…that had a lower detection limit…"
*Modified.*

Page 8, line 205.    Suggest"…lower detection limit of the NOx analyzer was…"
*Modified.*

Page 9, line 209.    Also concerning that the aerosol measurements were not made at the same heights at the HOM measurements.    What impact could this have?
*No significant impact, we only used aerosol measurements to calculate CS, and used CS to indicate air mass change in our case studies, which will not influence our conclusion. The aerosol particle lifetimes are on the order of days, and therefore minimal changes are expected between the two heights, as has been verified in earlier aerosol studies at the site.*

Page 9, line 216.    Suggest "…averaging intervals, except for the MT (in 1-hour averaging intervals)."
*Modified.*

Page 9, line 227-229.    Suggest "The mean air temperature and RH observed at ground level were…, and at the tower level were…".
*Modified.*

Page 9, line 230-231.   Suggest providing statistics for temperature, RH, and O3 separately for daytime and nighttime.
*Modified.*

Page 9, line 222 and page 10, line 223.   This statement is confusing.   The NOx detection limit is 50 pptv (line 205), so how does this relate to mean +/- standard deviation values given.   Need a bit more text to describe what was done statistically, and what the results say.
*The text was changed to:*

*"The $O_3$ concentrations measured at ground and tower levels were $21 \pm 8$ ppbv and $25 \pm 6$ ppbv, respectively.", and "…the mean NOx concentrations were mostly around the reported detection limit at $0.4 \pm 0.4$ ppbv (ground) and $0.4 \pm 0.5$ ppbv (tower), …"*

Page 20, line 235-236.   Suggest "…were generally higher than those above…"
*Modified.*

Page 11, line 242.   Since the transmission efficiencies are not used in the reduction of the data, this reviewer disagrees with the statement that the sum of the signals between m/z 200 to 600 represents the total HOM concentration.   This needs some reworking.   If the transmission efficiencies are not know, then suggest not giving HOM concentrations, but perhaps HOM signals.
*We had also considered this option, but ultimately concluded that since we know roughly the HOM concentrations, it would be more useful for a reader to compare actual concentrations, albeit they have high uncertainty. We tried to highlight this uncertainty in section 3.2. by writing:*

*"an uncertainty of -50%/+100%, was used in calculating the HOM concentrations for both instruments. Ultimately, the absolute HOM concentrations in this work are of secondary importance, as we focus on the relative comparison of HOM concentrations measured at different heights."*

*And*

*"In comparison to the direct determination of TE (Heinritzi et al., 2016), this method increases the uncertainty in the quantification of HOM concentrations. However, as mentioned, a more accurate knowledge of the exact HOM concentrations would not influence the main findings of this study."*

*We now try to even further emphasize this uncertainty by changing all the "total HOM concentration" to "estimated total HOM concentration", and adding below text to give a clearer statement in the manuscript:*

*"an uncertainty of at least -50%/+100%, was used in calculating the HOM concentrations for both instruments. Ultimately, the absolute HOM concentrations in this work are of secondary importance, as we focus on the relative comparison of HOM concentrations measured at different heights."*

Page 11, line 246-248. Are the statistics for all the data, or just daytime or nighttime? The value after +/- is presumably the standard deviation. This needs to be stated. In the last sentence "these differences" are mentioned, but it needs to be specifically stated which differences are being referred to (e.g. differences in the means above and below (fairly small), differences in the medians (larger), etc.)?
*The statistics were determined basing on the whole data. The value after the symbol "±" had been defined as (1σ standard deviation, page 9, line 230) in section 4.1. Additionally, there was an ~55% difference in mean values (~71% in median) between the two heights, which is quite large.*

*Now the sentence is modified to:*
*"The causes of these differences (~ 55% in mean and ~71% in median) frame the upcoming discussion."*

Page 11, line 252. Suggest adding statistics to demonstrate HOM concentrations at the two heights were not different during the day.
*Added. Now the sentence is:*

*"The total HOM concentrations at the two heights were not different during the day (mean ± 1σ standard deviation and median concentrations of $4.1 \pm 2.3 \times 10^8$ $cm^{-3}$ and $3.6 \times 10^8$ $cm^{-3}$ at ground level, $4.3 \pm 2.6 \times 10^8$ $cm^{-3}$ and $4.0 \times 10^8$ $cm^{-3}$ at tower level), which …"*

Page 11, line 254. Suggest a figure showing that the ratio (or some other metric) of the HOM concentrations at the two heights did not change with time during the day.
*Added, see Figure 8 and Figure 9.*

Page 11, line 257. Suggest including statistics and time dependence for the two heights (as above) for nighttime data.
*Added, see Figure 8 and Figure 9.*

Page 11, line 258. Suggest showing the temperature difference between the two heights in Figure 2 to help clearly show when there are temperature inversions.
*We have changed the temperature in Figure 2 from liner scale to log scale, which we think serves the same purpose.*

Page 11, line 261 and Figure 3. Suggest giving statistics to support the statement that there is good agreement around midday.
*Added. Now the sentence is:*

*"…representing the concentrations around noontime ($R^2 = 0.89$)…"*

Page 11, line 264 and Figure 3.   Suggest giving statistics for nighttime HOM concentrations to support the statement.
*Added. Now the sentence is:*

*"The points indicating the nighttime total HOM concentrations were scattered ($R^2 = 0.28$)…"*

Page 12, line 270.   Suggest "…shows the mean mass spectra…" and "…UMR, for m/z 200…" and remove "HOM measurements"
*Modified.*

Page 12, line 283.   Suggest "…strength and/or source-sink…"
*Modified.*

Page 14, line 292.   Suggest adding "likely" or "probably" in "…level are likely influenced by….".
*Modified.*

Page 14, line 294-296.   Suggest "…the potential impact of such micrometeorological phenomena on ground level HOMs, for the nights during the campaign without precipitation or instrument failure, were selected…"
*Modified.*

Page 14, line 297.   Suggest "…based on the occurrence of temperature inversions…"
*Modified.*

Page 14, line 299.   Suggest "…type category consisted of 6 nights...".
*Modified.*

Page 14, line 301.   Suggest "…lower than tower…"
*Modified.*

Page 14, line 307.   Suggest "…above the canopy was relatively…"
*Modified.*

Page 14, line 310.   Is it known that there are higher VOC emissions near the ground within forest canopies?   A reference or two would be good here.
*Rantala et al., (2014) is added to the text.*

Page 14, line 315.   Suggest "…similar in both categories and heights…"
*Modified.*

Page 16, Table 1.   This reviewer found the gray bars in the table (not the titles) confusing.   Suggest configuring the table differently.
*Modified.*

Page 17, line 344 and Table 2.   Were these categories done for all conditions, all times (looks like it is nighttime), and both heights?
*Yes, this table only included the nighttime data and both heights.*

Page 17, line 363.   Suggest "Roughly, Ri values in excess…" and "…stratified conditions appreciably…".
*Modified.*

Page 17, last paragraph and Figure 5.   Were these data for the ground level measurements?
*No, both ground and tower measurement data were involved in this paragraph and Figure 5, except for CS (determined based on the data measured at 8m a.g.l.) which was only used as an indicator of air mass change. The bulk Richardson number (Ri) is a scale of the air stability, and was calculated using the meteorology data of both ground and tower levels measurements. A detailed description was already given in section 4.3.2. We modify below texts to make a clearer statement in the manuscript:*

*"Figure 5b shows the time series of the trace gases, MT, and HOM groups of both ground and tower measurements during an "inversion night" case (September 8-9, from 21:00 to 03:00)."*

*And*

*"The parameters measured at tower level were not significantly affected by strong Ri fluctuations throughout the night, in contrast, significant variations were observed at ground level."*

*And the caption of Figure 5:*

*"Figure 1 (a) Time series of ground and tower concentrations of $CO_2$, $NO_x$, $O_3$, MT, and selected HOM groups in the selected "non-inversion night" (September 11), and (b) "inversion night" (September 8). Ri is calculated with the meteorology data of ground and tower levels. CS is determined based on the aerosol data measured at 8 m above ground level."*

Page 19, line 381.   Suggest "…significant decreases after midnight."
*Modified.*

Page 19, line 387.   Suggest "…HOM concentrations might…"

*Modified.*

Page 19, line 388.   Suggest "…but also due to some other processes such as additional losses."
*Modified.*

Page 19, line 391.   Suggest giving the location of the Alekseychik et al study.
*Modified. The study was conducted in the same SMEAR II station. Now the expression is:*

*"A previous study by Alekseychik et al., (2013) at SMEAR II station showed that…".*

Page 19, line 392.   Suggest "…Ri conditions in the…"
*Modified.*

age 19, line 396.   Suggest "…in significantly different O3…"
*Modified.*

Page 19, line 397.   See earlier comment about "sub-canopy".
*Modified.*

Page 20, top paragraph.   This reviewer found the use of T1, T2, etc confusing since capital T is usually reserved for temperature.   Suggest using different symbols.
*Modified. Now change to "N".*

Page 20, line 418.   Suggest "Note that these…"
*Modified.*

Page 20, line 421.   Suggest "area-to-volume ratio…".
*Modified.*

Page 20, line 424.   See earlier comment about "sub-canopy".
*Modified.*

Page 22, line 452.   Suggest "…however, such an analysis might only indicate the major…"
*Modified.*

Page 22, line 453.   Suggest "…holistic view of the entire mass spectrum…" or some other rewording.
*Modified.*

Page 23, line 461.   Suggest "…large differences could…"
*Modified.*

Page 23, line 463.    Suggest "…disappeared on the…"
*Modified.*

Page 23, line 468.    Suggest "…limitations still exist in this…"
*Modified.*

Page 23, line 472.    Suggest "…influence of horizontal advection could not be entirely ruled out as a contributor to..."
*Modified.*

Page 23, line 473.    Suggest "…HOM concentrations…".    Suggest rewording "largely changed species"
*Modified. Now the sentence is:*

*"… HOM concentrations (and other significantly changed species)"*

Page 23, line 473.    Suggest "…because of possible horizontal…"
*Modified.*

Page 23, line 477.    Suggest "…advection is probably minor…"
*Modified.*

Page 23, line 479.    Suggest "…evidence is still needed…"
*Modified.*

Page 23, line 479-280.    Suggest "…which highlights the need for…".    Also define "joint vertical-planar HOM studies".
*Modified, and now the sentence is:*

*"…which also highlights the need for joint vertical-planar HOM studies, measuring both vertical and horizontal distribution of HOM concentrations."*

Page 23, line 484.    Suggest "…IBAIRN campaign that took place in September 2016."
*Modified.*

Page 24, line 486.    Suggest "…that influence the abundance and trends of HOMs…"
*Modified.*

Page 24, lines 505-507.    Suggest "…close to the ground, and the effect of boundary layer…" and "…processes to HOM concentrations have…"
*Modified.*

Page 24, line 509.    Suggest "…HOM concentrations found in nocturnal inversion situations."
*Modified. Now the expression is:*

*"…reduced HOM concentrations in the inversion nights."*

Page 24, line 510.    Suggest "Influence of boundary…"
*Modified.*

Page 27, line 582.    There is a typo in this reference.
*Modified.*

References:

Alekseychik, P., Mammarella, I., Launiainen, S., Rannik, Ü. and Vesala, T.: Evolution of the nocturnal decoupled layer in a pine forest canopy, Agric. For. Meteorol., 174, 15–27, doi:10.1016/j.agrformet.2013.01.011, 2013.

Ehn, M., Thornton, J. A., Kleist, E., Sipilä, M., Junninen, H., Pullinen, I., Springer, M., Rubach, F., Tillmann, R., Lee, B., Lopez-Hilfiker, F., Andres, S., Acir, I.-H., Rissanen, M., Jokinen, T., Schobesberger, S., Kangasluoma, J., Kontkanen, J., Nieminen, T., Kurtén, T., Nielsen, L. B., Jørgensen, S., Kjaergaard, H. G., Canagaratna, M., Maso, M. D., Berndt, T., Petäjä, T., Wahner, A., Kerminen, V.-M., Kulmala, M., Worsnop, D. R., Wildt, J. and Mentel, T. F.: A large source of low-volatility secondary organic aerosol, Nature, 506(7489), 476–479, doi:10.1038/nature13032, 2014.

Jokinen, T., Sipilä, M., Richters, S., Kerminen, V.-M., Paasonen, P., Stratmann, F., Worsnop, D., Kulmala, M., Ehn, M., Herrmann, H. and Berndt, T.: Rapid Autoxidation Forms Highly Oxidized $RO_2$ Radicals in the Atmosphere, Angew. Chemie Int. Ed., 53(52), 14596–14600, doi:10.1002/anie.201408566, 2014.

Rantala, P., Taipale, R., Kajos, M. K., Patokoski, J., Ruuskanen, T. M., Rinne, J. and Aalto, J.: Continuous flux measurements of VOCs using PTR-MS — Reliability and feasibility of disjunct-eddy-covariance, surface-layer-gradient, and surface-layer-profile methods, Boreal Environ. Res., 19, 87–107, 2014.

Heinritzi, M., Simon, M., Steiner, G., Wagner, A. C., Kürten, A., Hansel, A. and Curtius, J.: Characterization of the mass-dependent transmission efficiency of a CIMS, Atmos. Meas. Tech., 9(4), 1449–1460, doi:10.5194/amt-9-1449-2016, 2016. Berndt, T., Scholz, W., Mentler, B., Fischer, L., Herrmann, H., Kulmala, M. and Hansel, A.: Accretion Product Formation from Self- and Cross-Reactions of $RO_2$ Radicals in the Atmosphere, Angew. Chemie Int. Ed., 57(14), 3820–3824, doi:10.1002/anie.201710989, 2018.

Anonymous Referee #2

The article by Zha et al. touches on important but challenging issues of how the extremely low volatility, highly oxidized molecules (HOMs aka ELVOC) behave below and above the tree canopy. The article is generally clear and makes many confident in- sights into the atmospheric variability of these molecules. Because these molecules have high propensity to form SOA, such measurements are needed and potentially valuable. However, I would have some important reservations to this version of the manuscript about the data interpretation and QC/QA, but hopefully, this can be successfully addressed by the authors.

**Major**
1) Two heights were chosen for measurements of HOMs by two different CI-APi-TOF instruments. Although I agree with the other reviewer that this might be regarded as a flaw, I think it does not have to be a critical flaw as long as there is a substantial effort to ensure that the two instruments were in perfect agreement. I am surprised why a portion of measurements was not conducted at the same height (either ground or tower) by both instruments side by side first before moving on to measuring at two heights. It is unfortunate because the collocated measurements would help to ensure that the data reported by two instruments are indeed identical. I am concerned that it might be difficult otherwise to demonstrate this, because there are numerous factors that may affect the agreement between two different instruments apart from the relative transmission. For example, fragmentation, clustering, declustering, or other processes modulated by changes in ambient temperature and humidity may have differed as a function of day, as a function of height, and over longer time scales in either or both instruments. It is therefore really difficult to get convinced if the differences are necessarily because of the height and not because of the differences in each instrument's quantification. A single point laboratory comparison of just the relative transmission of two instruments does not seem sufficient, because of the inability of assessing the factors which change over time. It would be ideal to calibrate regularly the instruments independently and assess separately the deviation from the theoretical transmission of the TOFs like in Heinritzi et al. (2016). Then you could compare the datasets and see how consistent they would be in the middle of a day and at night.
*We understand the reviewer's concern about the comparability of the two CI-APi-TOF mass spectrometers, and hope that we addressed this concern adequately in the response to reviewer 1. As a specific response here, the reviewer suggested temperature and humidity as two factors that might cause changes to the response of*

*the CI-APi-TOFs. We believe that Figures 8 and 9, and the corresponding text, that were added in the revised manuscript clearly show that the instruments agree well over a wide range of temperatures and humidities. The only times the instruments show a large discrepancy is when there is a temperature inversion and also key parameters that are known to influence HOM loadings (e.g. ozone) also show large deviations between the two heights.*

*Finally, if the reviewer's concern was validated, and small changes in ambient temperature or RH would cause the observed changes of ~one order of magnitude through changes in instrument response, it would call into question all published data from these instruments, not only our manuscript. The CI-APi-TOF mass spectrometer has been deployed in very different environments, such as forests, mountain tops, and coastal areas (Bianchi et al., 2016; Kürten et al., 2016; Sipilä et al., 2016; Yan et al., 2016), and no evidence for such erratic behavior has been suggested.*

2) Eq. 1 is only valid if the conditions in the reaction chamber or TOF chamber have not changed. Small variations in pressure, temperature or humidity could affect the calibration coefficient. It would be inappropriate to expect that if you set the collision rate the same for all the ions, the relative consistency in two instruments will be the same, because the sensitivity can change over time also because of the drift parameters and resulting issues such as different fragmentation which would not be accounted for by the single transmission correction. It is therefore recommended to take zeros and calibrations of the instruments frequently and none of these are shown except that permeation device is mentioned.

*We hope that most of these concerns were addressed in the new Figures 8 and 9. As also discussed in our earlier responses, HOM concentrations were just relatively quantified with Eq. 1, as we were not interested in the absolute HOM concentrations, but more cared about the comparability between the two CI-APi-TOF mass spectrometers. The relative transmission could be deduced from any noon-time, well-mixed period, and it stayed quite constant throughout the measurement period.*

*We also calculated the ambient pressure difference between ground and tower level, and find it unlikely that such small ambient pressure changes (variation was 4.05-4.28 hPa between ground and tower level) could result in ~20 times differences in HOM signals.*

3) I understand that the authors did not seem concerned about the absolute quantification (L165-166) but even the relative quantification is uncertain if one of the instrument was affected by different conditions or its sensitivity drifted throughout a day. It is unclear how the instruments were housed and if the temperature inside the

instruments were monitored and if it was consistent at the two heights. A visual schematic would be useful. One way around could be to either add a period of collocated measurements for a few days and compare the range of compounds or show how stable baselines, sensitivities, and transmissions, were throughout the measurement period if you did regular calibrations.

 *Most of these comments on inter-comparability have been addressed already in our responses. We added the following text in section 3.2 about how the instruments were housed and their working conditions:*
*"Both instruments were working in rooms with air-conditioning and room temperatures controlled at 25 ℃."*
*As also discussed earlier, although they would have been very useful. we can obviously not anymore produce any collocated measurements.*

4) Why do you assume a noon period should have identical HOMs concentrations at two heights? I thought these were the first measurements of HOMs at two heights or can you provide a reference? Again, I am surprised that this period was chosen for transmission comparison instead of collocating the two instruments for a longer period. I am not convinced that it is fair to assume that all the HOMs will have no concentration gradient during a day. I am worried that if the HOMs are lost more rapidly at the surface the relative transmission ratio between the instruments may be biased towards larger masses which could be expected to be lost as a function of mass-dependent volatility which could mimic the duty-cycle and transmission related mass discrimination.

*We agree with the reviewer that the vertical gradients of HOMs is likely not zero. However, we believe that it is a fair assumption that any such gradient would be small and its influence limited. During noon-time, there is very efficient vertical mixing taking place, and both the main sources (oxidants and VOCs) and sinks (aerosols) have very small gradients between the two measurement heights. While the ground and the canopy are also sink terms for HOM, the canopy is roughly mid-way between the measurement levels.*
*Additionally, Figure R1 shows the correlation between the normalized (to reagent ions) noontime tower and ground sulfuric acid (SA) signals (before correcting with relative TE) during the campaign. Due to SA's extremely low volatility and high diffusivity, it should have the largest gradient among all the measured compounds. However, they agreed very well and ground SA signals were only ~20% lower than tower signals, suggesting vertical gradient would not significantly affect our relative transmission curve. Moreover, vertical gradient would be the smallest during noontime and larger during night (because of the lower turbulence mixing), but good agreements were still found during the night of September 11 (Figure 8), between*

*ground and tower HOM measurements. Though the ratios were a bit higher in lower mass range (200-300 Th) during this non-inversion night, they were still much lower compared to September 8 (Figure 9, temperature inversion night), suggesting the large differences between ground and tower HOMs were not mainly from vertical gradient of HOMs.*

[Figure]

*Figure R1 Correlation between all the noontime (12:00 LT) ground and tower SA signals (1-hour averaged, normalized to reagent ions) during campaign.*

*Finally, the extent of the scatter in Fig. 1 is a clear indication that the governing parameter is indeed the molecular mass. While there is a correlation between molecular mass and volatility, the extent of the scatter in the plot would most likely increase dramatically if the volatility would be the governing parameter. Compare for example that a molecule with the same elemental composition, e.g. C10H16O8, can have isomers with orders of magnitude differences in vapor pressure (Kurtén et al., 2016). Therefore, the data clearly indicates that the mass-dependent transmission is more likely to explain the relative differences during noon-time than different HOM sink parameters.*

5) Fig. 2 shows a good agreement for HOMs during noon and not so good otherwise. This relative scaling seems rather arbitrary and not completely unsurprising given the data were normalized using Figure 1 which was derived for the noon time. I am skeptical if these differences really represent the effect of different heights. The differences are huge because the data are shown on a log scale. I would recommend showing the comparison on a linear scale.

*For the first part of the comment, we have addressed these issues earlier. As for the scale, since we are more interested in the relative difference between ground and*

*tower measurements and not the absolute HOM concentrations, we find it more reasonable to show the ground and tower HOM concentrations in log scale. The absolute differences are easy to read out from the graph, and we do not therefore think that there should be any risk for misinterpretation of the plot.*

6) Overall, I was missing a stronger link to chemical and physical properties of HOMs and a deeper insight into individual classes and not just the total sum of HOMs. In particular, the transport could be molecule specific and may not be unified across the full range of HOM vapor pressures. I feel that the dataset has a much higher potential for teaching a reader about the behavior of HOMs. The current version of the paper gives an impression of semiquantitative and speculative in terms of HOM vertical behavior. I am not convinced by looking at temperature variations (Fig. 2) that the inversion hypothesis is strong. There is much attention directed to the general remarks about advection and turbulent quantities which is difficult to infer how they affect HOMs without direct flux measurements of HOMs by eddy covariance.

*We certainly agree with the reviewer that this is a very rich data set and future work will look more into molecule-level differences. As we believe we have shown in these responses, the concentration differences are driven by decoupling of layer between the two measurement heights, but as is clear from the scatter in Fig. 9, the chemistry is changing drastically. But we also feel that it would go out of scope for this study to involve detailed discussions on the HOM chemistry in addition to all other topics covered. Additionally, we do show several different types of HOMs in Figure 5, and their main formation pathways were also listed in Table 2 in the manuscript.*

*The existence of temperature inversion in the boreal forest environment has been proven in many studies, and a small temperature inversion could result in decoupled layer formation and have significant influences to $O_3$, monoterpene and $CO_2$ concentrations (Rannik et al., 2009, 2012; Alekseychik et al., 2013; Chen et al., 2018). Since HOM concentrations have a strong dependence on $O_3$ and MT concentrations, it would inevitably be affected by temperature inversion. However, the potential importance of micrometeorology in HOM measurements had yet been recognized by most of the community, prompting us to publish our findings without further delays.*

**Minor**

7) I wonder why you are adopting the HOM nomenclature and not ELVOC. Mentel et al. (2015) suggested that the latter is more appropriate when referring to atmospheric impacts and HOMs if the focus is more on compounds' structure. As you are focusing on the behavior in the atmosphere I think considering volatilities could make sense.

Are you sure that all the molecules you report are highly oxidized (high carbon oxidation state)? That would not be true for compounds such as cyclic and linear siloxanes which should be subtracted from the HOMs class.

*In most studies where HOM or ELVOC have been used, they have been effectively defined in the same way, as the oxidized organic compounds that are detected by a nitrate ion based CI-APi-TOF (e.g.Ehn et al., 2014; Kirkby et al., 2016). However, according to a recent study by Kurtén et al., (2016), there is a large difference between ELVOC and HOM, and HOM measured with nitrate ion based CI-APi-TOF spans a wide range of volatilities (Kirkby et al., 2016), not only the ones with extremely low volatilities (ELVOC). Therefore, HOM is a better nomenclature for our study, since we can actually infer the amount of oxygen in the molecules, while assessing the volatility would require many assumptions. The reviewer also suggested that variations in volatility could cause differences in the expected gradients, but if all detected molecules were ELVOC, they would all behave identically. Note also that we use the term "highly oxygenated" and not "highly oxidized", and thus are not inferring an oxidation state, rather just the oxygen content.*

*There are also molecules in the studied mass range that are not very highly oxygenated, but the amount of these is small. This is a feature of our instruments, as the reagent ion ($NO_3^-$) of our CI-APi-TOF mass spectrometer is very selective and tends to charge molecules with high oxygen content (mostly with six or more O-atoms.*

8) Figure 3 does not provide much information when it is colored by the time of day. For the reasons mentioned above the scatter outside of the noon hour might not necessarily be because of less good mixing other than at noon. One could possibly learn more if the data were colored by potential temperature or relative humidity.

*We think this figure is quite important and needs to be colored by the time of day to show that noontime data is always well-correlated (which e.g. the reviewer questioned in comment 4). Since the relative transmission curve was determined based on the noontime data on September 9, Figure 3 shows that there were good agreements between ground and tower measurements during every day of the campaign, and large differences were usually observed during night.*

9) Figure 4c, the scatter looks weird that it is so much shifted (by an order of magnitude) but still reasonably correlated. I wonder if it is possible to evaluate any volatility dependent difference but it seems that it might be difficult if there is a high uncertainty in transmission differences.

*We are not sure if we interpret the reviewers comment correctly, but we think the reviewer may have interpreted the figure as a correlation between the timeseries of ground and tower HOMs, while it in fact shows the correlation between each ion in*

*the mean ground and tower spectra. In any case, since Figure 4b and 4c ultimately contain the same information as Figure 4a, we decided to remove these two figures from our manuscript completely.*

10) Figure 1 is technical and could be moved to SI. I would suggest to replace it by separate individual curves of theoretical and measured transmissions from each instrument and the datasets should be corrected individually.

*We believe Figure 1 is important for readers to understand the concept of this study, it is the basis to compare ground and tower HOM measurements. As also both reviewers' comments circled around this scaling, we feel it is critical to have it easily accessible in the main text. We also cannot replace it with any measured transmission curves as the reviewer suggests, since such do not exist. Similarly, we are not aware of any method to derive theoretical transmission curves as such would need to take into account all dimensions, flows, the ~30 voltages and the two radiofrequencies inside the APi-TOF. Therefore our only option remains to do the scaling as shown in Fig. 1, which we believe we have validated in our responses and the new version of the manuscript.*

**Technical**

L67 space between ")" and "from".
*Corrected.*

Table 1, ensure the number of significant figures is consistent and as appropriate.
*Corrected.*

Table 2, if there is only one N atom in a molecule there is no need to add 1. For example, $C_{10}H_{15}O_{11}N_1$ should be $C_{10}H_{15}O_{11}N$.
*Corrected.*

References:
Alekseychik, P., Mammarella, I., Launiainen, S., Rannik, Ü. and Vesala, T.: Evolution of the nocturnal decoupled layer in a pine forest canopy, Agric. For. Meteorol., 174, 15–27, doi:10.1016/j.agrformet.2013.01.011, 2013.

Bianchi, F., Tröstl, J., Junninen, H., Frege, C., Henne, S., Hoyle, C. R., Molteni, U., Herrmann, E., Adamov, A., Bukowiecki, N., Chen, X., Duplissy, J., Gysel, M., Hutterli, M., Kangasluoma, J., Kontkanen, J., Kürten, A., Manninen, H. E., Münch, S., Peräkylä, O., Petäjä, T., Rondo, L., Williamson, C., Weingartner, E., Curtius, J., Worsnop, D. R., Kulmala, M., Dommen, J. and Baltensperger, U.: New particle

formation in the free troposphere: A question of chemistry and timing., Science, 352(6289), 1109–12, doi:10.1126/science.aad5456, 2016.

Chen, X., Quéléver, L. L. J., Fung, P. L., Kesti, J., Rissanen, M. P., Bäck, J., Keronen, P., Junninen, H., Petäjä, T., Kerminen, V.-M. and Kulmala, M.: Observations of ozone depletion events in a Finnish boreal forest, Atmos. Chem. Phys., 18(1), 49–63, doi:10.5194/acp-18-49-2018, 2018.

Ehn, M., Thornton, J. A., Kleist, E., Sipilä, M., Junninen, H., Pullinen, I., Springer, M., Rubach, F., Tillmann, R., Lee, B., Lopez-Hilfiker, F., Andres, S., Acir, I.-H., Rissanen, M., Jokinen, T., Schobesberger, S., Kangasluoma, J., Kontkanen, J., Nieminen, T., Kurtén, T., Nielsen, L. B., Jørgensen, S., Kjaergaard, H. G., Canagaratna, M., Maso, M. D., Berndt, T., Petäjä, T., Wahner, A., Kerminen, V.-M., Kulmala, M., Worsnop, D. R., Wildt, J. and Mentel, T. F.: A large source of low-volatility secondary organic aerosol, Nature, 506(7489), 476–479, doi:10.1038/nature13032, 2014.

Kirkby, J., Duplissy, J., Sengupta, K., Frege, C., Gordon, H., Williamson, C., Heinritzi, M., Simon, M., Yan, C., Almeida, J., Tröstl, J., Nieminen, T., Ortega, I. K., Wagner, R., Adamov, A., Amorim, A., Bernhammer, A.-K., Bianchi, F., Breitenlechner, M., Brilke, S., Chen, X., Craven, J., Dias, A., Ehrhart, S., Flagan, R. C., Franchin, A., Fuchs, C., Guida, R., Hakala, J., Hoyle, C. R., Jokinen, T., Junninen, H., Kangasluoma, J., Kim, J., Krapf, M., Kürten, A., Laaksonen, A., Lehtipalo, K., Makhmutov, V., Mathot, S., Molteni, U., Onnela, A., Peräkylä, O., Piel, F., Petäjä, T., Praplan, A. P., Pringle, K., Rap, A., Richards, N. A. D., Riipinen, I., Rissanen, M. P., Rondo, L., Sarnela, N., Schobesberger, S., Scott, C. E., Seinfeld, J. H., Sipilä, M., Steiner, G., Stozhkov, Y., Stratmann, F., Tomé, A., Virtanen, A., Vogel, A. L., Wagner, A. C., Wagner, P. E., Weingartner, E., Wimmer, D., Winkler, P. M., Ye, P., Zhang, X., Hansel, A., Dommen, J., Donahue, N. M., Worsnop, D. R., Baltensperger, U., Kulmala, M., Carslaw, K. S. and Curtius, J.: Ion-induced nucleation of pure biogenic particles, Nature, 533(7604), 521–526, doi:10.1038/nature17953, 2016.

Kürten, A., Bergen, A., Heinritzi, M., Leiminger, M., Lorenz, V., Piel, F., Simon, M., Sitals, R., Wagner, A. C. and Curtius, J.: Observation of new particle formation and measurement of sulfuric acid, ammonia, amines and highly oxidized organic molecules at a rural site in central Germany, Atmos. Chem. Phys, 16, 12793–12813, doi:10.5194/acp-16-12793-2016, 2016.

Kurtén, T., Tiusanen, K., Roldin, P., Rissanen, M., Luy, J.-N., Boy, M., Ehn, M. and Donahue, N.: α-Pinene Autoxidation Products May Not Have Extremely Low Saturation Vapor Pressures Despite High O:C Ratios, J. Phys. Chem. A, 120(16), 2569–2582, doi:10.1021/acs.jpca.6b02196, 2016.

Rannik, Ü., Mammarella, I., Keronen, P. and Vesala, T.: Vertical advection and

nocturnal deposition of ozone over a boreal pine forest, Atmos. Chem. Phys., 9(6), 2089–2095, doi:10.5194/acp-9-2089-2009, 2009.

Rannik, Ü., Altimir, N., Mammarella, I., Bäck, J., Rinne, J., Ruuskanen, T. M., Hari, P., Vesala, T. and Kulmala, M.: Ozone deposition into a boreal forest over a decade of observations: evaluating deposition partitioning and driving variables, Atmos. Chem. Phys., 12(24), 12165–12182, doi:10.5194/acp-12-12165-2012, 2012.

Sipilä, M., Sarnela, N., Jokinen, T., Henschel, H., Junninen, H., Kontkanen, J., Richters, S., Kangasluoma, J., Franchin, A., Peräkylä, O., Rissanen, M. P., Ehn, M., Vehkamäki, H., Kurten, T., Berndt, T., Petäjä, T., Worsnop, D., Ceburnis, D., Kerminen, V.-M., Kulmala, M. and O'Dowd, C.: Molecular-scale evidence of aerosol particle formation via sequential addition of HIO3, Nature, 537(7621), 532–534, doi:10.1038/nature19314, 2016.

Yan, C., Nie, W., Äijälä, M., Rissanen, M. P., Canagaratna, M. R., Massoli, P., Junninen, H., Jokinen, T., Sarnela, N., Häme, S. A. K., Schobesberger, S., Canonaco, F., Yao, L., Prévôt, A. S. H., Petäjä, T., Kulmala, M., Sipilä, M., Worsnop, D. R. and Ehn, M.: Source characterization of highly oxidized multifunctional compounds in a boreal forest environment using positive matrix factorization, Atmos. Chem. Phys., 16(19), 12715–12731, doi:10.5194/acp-16-12715-2016, 2016.

---

## Referee Report (RR1)

Review of "Vertical characterization of Highly Oxygenated Molecules (HOMs) below and above a boreal forest canopy" by Zha et al.

This paper describes results of measurements of HOMs near the surface from atop a tower using mass spectroscopic instrumentation. Of particular focus are the measured differences at night when temperature inversions were present or absent. The results presented are interesting and provide useful insight into the nature of HOMs produced in the boreal forest at night, and complement the many other papers on HOMs in the literature.

This revised version of the paper satisfactorily addresses this reviewer's concerns in the first version. This paper is suitable for publication, but suggest a thorough check of the English before final submittal as there are some minor issues there.

---

## Author Response (AR2)

*We thank the editor for the additional comments. The editor's comments are listed below and are followed by our replies (in italics).*

Comments to the Author:

I have two major concerns on this manuscript and encourage the authors to elaborate more about the points below.

1. The major concern, as both reviewers address, is the instrumental or sampling biases due to the missing inter-comparison and horizontal separation in their set-up, during the mission. I do understand the limitations in real mission, so I suggest the authors to conduct more lab tests to address the relative TE (transmission efficiency) dependence of two instruments on various meteorological parameters; repeat figure 1 type experiments for different parameters such as temperature, pressure and RH to verify the possible biases in instruments in quantitative way. (The ionization efficiency may vary with that kinds of parameters.).

*In order to address the editor's concern about the potential biases due to the missing inter-comparison and horizontal separation, we now performed a side-by-side inter-comparison at the site from June 15 to 21, 2018. The "tower" CI-APi-TOF was deployed to the container where ground measurement was conducted and measured together with the "ground" CI-APi-TOF for 6 days. Figure R1 shows ambient temperature, RH, and the total HOM concentrations measured by the "ground" and "tower" CI-APi-TOFs during the inter-comparison. Clear variations of ground temperature (from 6.6 ℃ to 24.5 ℃) and RH (27 % to 100 %) were observed, but a good agreement ($R^2$=0.91, as shown in Figure R2) could still be found between the concentrations obtained from the two instruments, which confirmed that the two CI-APi-TOFs are comparable, and their consistency is not biased by ambient temperature or RH change. The "ground" total HOM concentrations in Figure R1 were not corrected by the relative TE (transmission efficiency) curve, thus in Figure R1 there is a nearly constant ratio (a factor of 6, as also shown in Figure R2) between the concentrations from "ground" and "tower" instruments. This difference is in good accordance with the TE curve shown in Figure 1 in our manuscript.*

[Figure]

*Figure R1 Time series of RH, temperature, and total HOM concentration measured by the "ground" and "tower" CI-APi-TOFs during the side-by-side inter-comparison from June 15 to 21, 2018.*

[Figure]

*Figure R2 Correlation between total HOM concentrations measured by "ground" and "tower" CI-APi-TOFs (red circles), and correlation after multiplying concentration from "ground" instrument by a factor of 6 (blue circles).*

*For comparison, Figure R3 shows ambient measurement data from May 24 to 30, 2018, shortly before the side-by-side inter-comparison. The most significant nighttime temperature inversion during this period was observed in the night of May 24, with a maximum ground and tower temperature difference of -6.5 ℃. The tower total HOM concentrations were generally stable or increased slightly during the night. In contrast, the ground concentrations kept decreasing until temperature inversion disappeared, after which the instruments again track each other closely. Similar opposite patterns between the two instruments were also observed in other inversion nights during this week, all analogous to our measurements in 2016.*

[Figure]

*Figure R3 Time series of RH, temperature, and total HOM concentrations at ground and tower levels during field measurement from May 24 to 30, 2018. Nights with temperature inversion are framed with black rectangles.*

*Furthermore, there is no indication from both experimental and theoretical studies shows that the large differences observed during inversion nights in our study were due to the changes in the instrumental response to changing ambient parameters, such as RH, temperature, and pressure:*

1. *RH: In the chemical ionization mass spectrometry community, it has long been known that reagent ions (such as nitrate) that form dimers with their neutral precursor (nor nitrate: $HNO_3*NO_3^-$) are much less impacted by humidity than reagent ions that do not form dimers (e.g. $I^-$, $Br^-$). This was recently concluded specifically for HOM utilizing a completely theoretical approach (Hyttinen et al., 2018) For the case of iodide CIMS (CI-APi-TOF), Lee et al. (2014) tested the humidity dependence and the results show that the change in sensitivity under any reasonable ambient conditions is less than a factor 2. Considering that iodide is going to be much more sensitive to humidity changes than the nitrate-based CI-APi-TOF, we argue that the nearly order-of-magnitude changes in HOM concentrations that we observed in our study are highly unlikely to be caused by changes in instrument sensitivity due to changes in RH.*

2. *Temperature: Both instruments are housed in temperature-controlled containers, and as such, even though ambient temperature changes, the instrument temperature does not. The sampled air (7 LPM) is mixed with a larger sheath flow (35 LPM, at room temperature), which severely dampens any changes in sample flow temperatures already in the inlet. This is of course also the case for RH in point 1.*

3. *Pressure: The temporal pressure variation during our study period was on the order of one percent. The difference in pressure between ground and tower, on the other hand, based on the hydrostatic equation, is around 5mbar, i.e 0.5 %. This value itself will have varied only a negligible amount. These pressure changes are very small, and we are not aware of any physical process that could cause a measurable change in our instrumental response due to these variations. The charging of most HOM is governed by the collision rate of HOM molecules with the charging ions (Ehn et al., 2014), as the formed $HOM*NO_3^-$ clusters are thermodynamically extremely stable (Hyttinen et al., 2015).*

*We sincerely hope that our new measurements together with the above argumentation adequately address all of the editor's concerns on this subject.*

2. If the measurements were conducted within the roughness sublayer, point measurement likely have limitations in inferring larger scale exchange (Raupach, M. R. 1987). More cautious descriptions are required for the implication part of this manuscript unless the authors describe more about the turbulent characteristic at the site.

*We make below modifications to the manuscript to have a more cautious description of the implication.*

*In the abstract part, we add:*

*"However, much lower HOM concentrations were frequently observed at ground level, which was likely due to the formation of a shallow decoupled layer below the canopy."*

*AND*

*"Our findings also illustrate that near-ground HOM measurements conducted in strong stably stratified conditions at this site might only be representative of a small fraction of the entire nocturnal boundary layer."*

*In the introduction part, we modify below text from:*

*"...the HOMs at these different heights are explicitly analyzed and characterized in conjunction with auxiliary turbulence and micrometeorological measurements."*

*To*

*"...the HOMs at these different heights at SMEAR II station are analyzed and characterized in conjunction with auxiliary turbulence and micrometeorological measurements."*

*In the conclusion/implication part, we add:*

*"..., showing that ground-based HOM measurement at this site might not be representative for the entire nocturnal boundary layer."*

*AND*

*", and the possible effect of boundary layer dynamics and micrometeorological processes to the HOM concentrations have rarely been considered"*

*A sentence is added to point out that the implication is still limited by the current experiment design:*

*"However, there are still limitations due to current experiment design, such as horizontal separation in instrument set-up, or the uncertainties from using point measurements at two heights to infer larger scale exchange."*

*We also add below text to Section 3.1 to describe more details about turbulent characteristics at the measurement site:*

*"...a total leaf area index (LAI) of ~6.5 $m^2m^{-2}$, a stand density of ~1400 trees $ha^{-1}$, and an average diameter at breast height (DBH) of ~0.16 m (Bäck et al., 2012; Launiainen et al., 2013). The forest floor is majorly covered with a shallow dwarf shrub (a LAI of ~0.5 $m^2m^{-2}$) and moss layer (a LAI of ~1 $m^2m^{-2}$) (Kulmala et al., 2008; Launiainen et al., 2013)."*

Minor comments

Error or uncertainty analysis on the gradient of HOM would be helpful to convince the reader; more scatter on higher m/z in figure 1 may also derive the large variation in higher m/z in figure 8 and 9. Quantitative description about HOM difference would be plausible to proof the existence of HOM.

*We are not sure if we interpret the editor's comments correctly but have now added discussion on the uncertainties in the HOM concentration estimates. However, concerning the last sentence, we do not feel that the existence of HOM is beyond any doubt at this stage, considering the wealth of studies on the topic.*

*In order to determine uncertainties in concentration estimations between the two instruments, and to account for error related to the assumption of strong mixing leading to equal concentrations at the tower and ground during daytime, we compared the sulfuric acid signals between the two instruments. Figure S1 (now added to the SI) shows scatter plots between ground and tower sulfuric acid (SA) signals from 08:00 to 16:00 LT. During nighttime (before 07:00 and after 17:00), sulfuric acid concentrations were very low, sometimes below detection limit, and therefore were not included in this figure. At 08:00 and 16:00 LT, vertical turbulent mixing was still weak, and ground and tower experiments might be experiencing different turbulent mixing strength, thus lower coefficients of determination were obtained ($R^2$=0.35 at 08:00 LT, and 0.86 at 16:00 LT). However, from 09:00 to 15:00 LT, a uniform (well-mixed) boundary layer condition occurred. Strong correlations (all $R^2$ were higher than 0.97) were found between the two heights, and an average instrumental error of ~26% could be obtained based on all the measured tower/ground sulfuric acid ratios during this period.*

[Figure]

*Figure S1 Correlation between ground and tower sulfuric acid signals from 08:00 – 16:00 LT (1-hour averaged, normalized to reagent ions) during campaign. The black line denotes 1:1 ratio.*

*Next, we calculated the error associated with the TE correction. An average error of ~10% was determined from the higher mass range (m/z 500 - 600) in Figure 1, which could be considered as the upper limit of the error from our regression. From these two errors (instrumental and regression), we can calculate an estimate of the total error using the method of error propagation. The resulting value is ~28%, which can be compared to the two case studies in Figure 5. The average total HOM concentrations during the non-inversion night differed by (total HOM in tower) / (total HOM at ground) = ~20% while during the inversion night they differed by ~265%. While 20% is within/close to the calculated uncertainty of 28%, the difference during the inversion night (265%) is clearly larger and significant.*

*Below text is added to Section 3.2 to describe the uncertainty from HOM vertical gradient in a more quantitative way:*

*"To test our assumption of negligible vertical gradients of HOMs during daytime, we analyzed the behavior of sulfuric acid. We found that the uncertainty related to this assumption*

*corresponds to a value of 26% (see Figure S1). An upper limit of uncertainty relating to our TE correction (Figure 1) was also estimated, yielding a value of 10%, giving a total uncertainty from these two sources of 28%. This value is much smaller than the observed deviation of HOM concentrations during inversion nights (e.g. Figure 5)."*

2. Address the physical meaning of curve fitting in figure 1.

*The physical meaning of the curve fitting is to make the two instruments directly comparable, by removing differences in the mass-dependent transmission of ions through the two CI-APi-TOF instruments. The transmission of a certain instrument depends on multiple factors, including the detailed alignment of all part inside the instrument, exact pressures at each of the different stages, voltage settings, etc. For a thorough description of different parameters influencing the ion transmission, we refer to section 2 "Mass discrimination effects" in Heinritzi et al. (2016).*

*As stated by Heinritzi et al. (2016), it is basically impossible to calculate the transmission efficiency, and in practice it needs to be experimentally determined. For determining exact concentrations, this needs to be done through fairly tedious calibrations. In our case, where we only compare two instruments relative to each other, it was sufficient to scale the instruments to each other during periods when they were measuring the same sample (i.e. the turbulent, noon-time period).*

*To better clarify the physical meaning of the fitted curve, we add below texts to Section 3.2:*

*"For a detailed discussion on factors affecting the TE of a CI-APi-TOF, we refer to Heinritzi et al. (2016)."*

*AND*

*"A fitted relative TE curve ($R^2 = 0.97$), which represents how the TE of the tower CI-APi-TOF was changed at each m/z over the TE of the ground one, was obtained using power law regression."*

3. Any temperature, RH, pressure dependence in 1:1 correlation between tower and ground HOM? (I mean figure 3 colored by different variables.)

*As shown in Figure R4, points with different colors can be seen on the 1:1 line in each subplot, indicating that the vertical distribution of noontime HOM concentrations were not strongly depending on temperature, RH, or air pressure. Monoterpene emissions (which are HOM precursors) have a strong temperature dependence, and therefore the higher temperature show higher HOM concentrations (at both heights).*

[Figure]

*Figure R4 Correlations between ground and tower HOM measurements. The black lines denote 1:1 ratio. Color codes indicate temperature (left), RH (middle) and pressure (right) measured at ground (lower row) and tower (upper row) levels.*

4. If the authors put the information in Table 1 as box whisker plots, that would be easy to read

*We agree that there were too many numbers in Table 1. As the main point of the table was to show the significant difference between non-inversion and inversion nights, we realize that the mean and median values are already capable in doing that. Thus, we delete the rows of 25/75 percentile data in Table 1, to simplify the table and make it easier to read.*

5. Brief mention and more discussion about condensation sink would be helpful to the readers who are not in this field.

*Below texts are added to the Section 4.3.2:*

*"A rapid decrease was found in CS, which represents the rate of condensation of low-volatile vapors onto the existing aerosol particles (Dada et al., 2017), implying that the aerosol population was also altered."*

*AND*

*"However, all the HOM groups showed significant decrease after midnight, despite the CS (generally the main sink for HOM in the atmosphere) staying practically constant."*

6. Difference or ratio plot between tower and ground for figure 7 would be more informative; current plot has many overlaps in points. This part may have a chance to link with figure 8 and 9 for deeper analysis than descriptive results.

*Modified, Figure 7 is changed to:*

*"*

[Figure]

*Figure 7 Mass defect (MD) plots of the selected (a) "non-inversion night" case (September 11); and (b) "inversion night" case (September 8). Color code indicates the ratios between tower/ground HOM concentrations. Grey shade area denotes the dimer range (m/z 450-600).*

*"*

*And we also modify below texts from:*

*"The mass defect (MD) plot, with the exact masses of the compounds on the x-axis, the deviation from the integer mass on the y-axis, the compounds plotted in circles and the areas scaled by concentrations, shows the abundance and chemical speciation of all the detected HOMs in the spectra. Figure 7a and 7b are MD plots showing the mean spectra of the selected non-inversion night (September 11) at tower and ground levels. Without the formation of a decoupled layer, nearly identical composition distribution of HOMs were observed. In contrast, during the inversion night (September 8, Figure 7c and 7d), large differences could be found between the two measurement heights."*

*To*

*"The mass defect (MD) plot (Figure 7) separates all identified compounds according to their exact masses on the x-axis and the deviation from the integer mass on the y-axis. Each circle represents a compound, with the areas scaled by concentrations, and colored by the ratios between tower and ground concentrations. Figure 7a and 7b are MD plots showing the mean spectra of the selected non-inversion night (September 11) and inversion night (September 8).*

*Without the formation of a decoupled layer, nearly the same concentration distributions of HOMs were observed. In contrast, during the inversion night (September 8, Figure7b), large differences could be found between the two measurement heights."*

7. If possible, I suggest the authors to add figure R1 for all time windows of a day in supplement as a way to prove the gradient in HOM is real.

*Added, see Figure S1 in previous response.*

8. I feel the description about the forest are missed in this manuscript especially for the readers who are not familiar with the site. Description about the canopy structure parameters (i.e. LAI, leaf distribution, tree species etc.) are required not only for the turbulent characteristics but also for the sources and sinks of HOM. I also suggest the authors to add a schematic of the site including the information about all the measurements which took place at the site to ease the reader for that information.

*A description of forest structure and other parameters were added, see previous response. Below schematic figure of the site was added to the supplementary file as Figure S2.*

[Figure]

*Figure S2 Map from Google Earth showing locations of all the measurements during IBAIRN campaign at SMEAR II station.*

*We also add below text in Section 3.2:*

*"A schematic figure showing locations of all the measured parameters used in this study is provided in Figure S2."*

---

## Author Response (AR3)

[revised manuscript text omitted]